# The silicon cycle impacted by past ice sheets

Jon R. Hawkings[1], Jade E. Hatton [1], Katharine R. Hendry [2], Gregory F. de Souza[3], Jemma L. Wadham[1], Ruza Ivanovic [4], Tyler J. Kohler[5], Marek Stibal[5], Alexander Beaton[6], Guillaume Lamarche-Gagnon[1], Andrew Tedstone[1], Mathis P. Hain [7,8], Elizabeth Bagshaw[9], Jennifer Pike [9] & Martyn Tranter [1]

Globally averaged riverine silicon (Si) concentrations and isotope composition ($\delta^{30}$Si) may be affected by the expansion and retreat of large ice sheets during glacial−interglacial cycles. Here we provide evidence of this based on the $\delta^{30}$Si composition of meltwater runoff from a Greenland Ice Sheet catchment. Glacier runoff has the lightest $\delta^{30}$Si measured in running waters (−0.25 ± 0.12‰), significantly lower than nonglacial rivers (1.25 ± 0.68‰), such that the overall decline in glacial runoff since the Last Glacial Maximum (LGM) may explain 0.06–0.17‰ of the observed ocean $\delta^{30}$Si rise (0.5–1.0‰). A marine sediment core proximal to Iceland provides further evidence for transient, low-$\delta^{30}$Si meltwater pulses during glacial termination. Diatom Si uptake during the LGM was likely similar to present day due to an expanded Si inventory, which raises the possibility of a feedback between ice sheet expansion, enhanced Si export to the ocean and reduced $CO_2$ concentration in the atmosphere, because of the importance of diatoms in the biological carbon pump.

[1] Bristol Glaciology Centre, School of Geographical Sciences, University Road, Bristol BS8 1SS, UK. [2] School of Earth Sciences, University of Bristol, Bristol BS8 1RJ, UK. [3] Institute of Geochemistry and Petrology, ETH Zurich, Clausiusstrasse 25, 8092 Zürich, Switzerland. [4] School of Earth and Environment, University of Leeds, Leeds LS2 9JT, UK. [5] Department of Ecology, Charles University, Viničná 7, 12844 Prague 2, Czech Republic. [6] National Oceanography Centre, University of Southampton Waterfront Campus, European Way, Southampton SO14 3ZH, UK. [7] Earth and Planetary Sciences, University of California, Santa Cruz, CA 95064, USA. [8] Ocean and Earth Science, National Oceanography Centre Southampton, University of Southampton Waterfront Campus, European Way, Southampton SO14 3ZH, UK. [9] School of Earth and Ocean Sciences, Cardiff University, Main Building, Park Place, Cardiff CF10 3AT, UK. Correspondence and requests for materials should be addressed to J.R.H. (email: jon.hawkings@bristol.ac.uk)

Silicon (Si) plays a crucial role in global biogeochemical cycles because it is an essential nutrient for a number of marine organisms, including some species of sponges, radiolarians, silicoflagellates, and diatoms[1]. Marine diatoms account for 35–70% of marine primary production[2], and are therefore key in maintaining ecosystem health, the ocean biological pump, and atmospheric carbon fixation[3]. The input of Si from terrestrial weathering via rivers is crucial as it sustains diatom productivity over the ocean's Si residence time[1]. Thus, understanding the sensitivity of the Si cycle in the past, and its likely response to future climate warming, is important for marine ecosystem change, biogeochemical carbon cycling, and by association the efficiency of the ocean's biological carbon pump.

Variations in the $\delta^{30}Si$ of natural waters, sediments, and biogenic silica reflect fractionation during continental and oceanic biogeochemical processing[4]. Lighter isotopes are incorporated into solids, for example during the precipitation of secondary weathering products and diatom frustule formation, thus inducing fractionation from parent material values and usually leading to accumulation of heavier isotopes in the dissolved Si phase[4,5]. The $\delta^{30}Si$ of biogenic silica in marine sediment cores has been used as a proxy to explore past oceanic dissolved silica concentrations (DSi)[6], infer diatom utilisation of Si[7,8], and investigate changes in Si source inputs[4]. There has been a focus on the $\delta^{30}Si$ change from the Last Glacial Maximum (LGM; ~21,000 years ago) to the present day, with Southern Ocean marine biogenic opal records showing a shift in $\delta^{30}Si$ of ~+0.5–1.0‰, according to Southern Ocean core records[7,8]. This has been explained by changes in dissolved silica utilisation in surface waters[7], variation in terrestrial silica inputs[4,9] and by oscillations in intermediate and deep-water DSi as a result of changes in oceanic circulation[8]. Until recently, modelling studies have assumed that riverine $\delta^{30}Si$ input was uniform over glacial−interglacial timescales in their first-order interpretation of downcore diatom records[4,10,11]. However, past research indicates that this is unlikely, and that at least part of the $\delta^{30}Si$ shift from the LGM to the present day can be explained by a change in the $\delta^{30}Si$ and/or magnitude of input fluxes, due to temporal changes in terrestrial weathering regimes[4,9,12,13]. There is additional uncertainty in interpretation of palaeo-records as downcore biogenic opal isotopic composition are likely to be a complex mixture of changes in silicic acid utilisation and concentrations, the $\delta^{30}Si$ of the whole ocean silicon isotope inventory, and more localised changes in inputs. This is problematic given most core records come from the Southern Ocean at present. The role of the changing extent of ice sheets (i.e. palaeo-ice sheets, PIS) since the LGM on the Si cycle has yet to be fully considered[14], despite their known impact on the global hydrological cycle[15] and weathering of continental rocks[16].

Glaciers and ice sheets covered nearly 30% of the Earth's land surface at their greatest extent during the LGM, including much of North America and northwestern Eurasia[17]. Melting of this ice during deglaciation raised sea levels by ~130 m and exported large quantities of eroded sediment into the oceans[18,19]. The last deglaciation contained two major, rapid ice melt events: Meltwater Pulse 1a (MWP1a; ~14,000–15,000 BP) where sea levels rose by 12–22 m in <350 years[20] and Meltwater Pulse 1b (MWP1b; ~11,000 BP) where sea levels may have risen by up to 10 m in ~500 years[21]. Glaciers and ice sheets are dynamic components of regional nutrient cycles[22,23], exporting significant quantities of dissolved[24] and labile amorphous silica (ASi) attached to fine-grained glacial suspended particulate matter (SPM)[14], which are likely to impact primary productivity in surrounding oceans[25]. Large silica fluxes from glaciated regions likely lead to preferential growth of diatoms in downstream marine ecosystems compared to other nonsilicifying phytoplankton species[24]. However, silica fluxes and their associated $\delta^{30}Si$ signature from the PIS have been overlooked in studies of ancient Si cycling[26–28], despite evidence suggesting the role of terrigenous sediment delivery to the ocean is underappreciated in global elemental cycles[29]. Thus far it has been found that glacial rivers in Iceland have a distinctive low $\delta^{30}Si$[12,13], but these data are based on point measurements rather than seasonal time series, and no data exist for large glacial systems more representative of PIS.

Here, we present a unique time series of DSi and ASi concentrations and associated $\delta^{30}Si$ composition for subglacial meltwaters exiting Leverett Glacier, a large (~600 km$^2$)[30] catchment of the Greenland Ice Sheet (GrIS). These data are used as a modern-day analogue for PIS runoff to investigate its potential influence on the Si cycle over glacial−interglacial and millennial timescales. Our findings suggest changing glacial silica fluxes could explain roughly 10–30% of the silicon isotope variation recorded in siliceous organisms since the LGM, instead of previously invoked changes in marine biological productivity or ocean circulation. Ice sheets are likely to have delivered large quantities of isotopically light silica to the oceans during periods of greater glacial activity, thereby augmenting the ocean's Si inventory and sustaining diatom productivity.

## Results

**Meltwater sampling.** Samples were collected from the proglacial river, ~1 km downstream from where subglacial meltwater exits the GrIS, from early May through July 2015. Leverett Glacier (67.06° N 50.15° W; Fig. 1) has a mean ablation season (May −September) discharge of ~150 m$^3$ s$^{-1}$ (Fig. 2, Supplementary Fig. 1), and is an order of magnitude larger than other glaciers studied for Si concentrations (DSi and ASi) and corresponding $\delta^{30}Si$ to date. The bedrock geology is predominantly Precambrian Shield gneiss/granite, broadly similar to much of northern Canada and Scandinavia, covered by the Eurasian and North American ice sheets[31] (Fig. 3). We contend that Leverett Glacier provides a first-order modern-day analogue for the PIS due to its size and bedrock type[31].

**Dissolved and amorphous silica concentrations in meltwaters.** DSi and ASi concentrations in Leverett Glacier runoff in 2015 were consistent with those previously reported for the GrIS[14,24]. Discharge-weighted mean concentrations were 20.8 (9.2–56.9) μM for DSi and 229 (69.8–336.6) μM for ASi. This equates to estimated DSi and ASi catchment fluxes of 30 (13–83) Mmol year$^{-1}$ and 331 (101–488) Mmol year$^{-1}$, within the same range as those reported at Leverett Glacier for the 2009–2012 period (396–1575 Mmol year$^{-1}$)[22]. Meltwater outburst events (shaded red in Fig. 2) in response to supraglacial meltwater forcing of the subglacial system[32] promote elevated DSi and ASi concentrations as Si-rich stored waters and sediments are flushed from the ice sheet bed.

**Si isotope composition of dissolved and amorphous Si.** We collected the first measurements of SPM $\delta^{30}ASi$ ($\delta^{30}Si$ of ASi; discharge-weighted mean of $-0.21 \pm 0.06$‰, $n = 11$), extracted using a weak alkaline leach (see Methods and SI). Our $\delta^{30}ASi$ values are lighter than those of local bedrock collected near the sampling site ($0.00 \pm 0.07$‰; $n = 3$) and bulk suspended sediment $\delta^{30}Si$ ($-0.09 \pm 0.07$‰; $n = 4$; grey-shaded region in Fig. 2d) by ~0.1–0.2‰. The discharge-weighted mean for $\delta^{30}DSi$ ($\delta^{30}Si$ of DSi) is extremely light, at $-0.25 \pm 0.12$‰ ($n = 16$), which is similar to $\delta^{30}ASi$, but significantly lower than the bedrock and SPM.

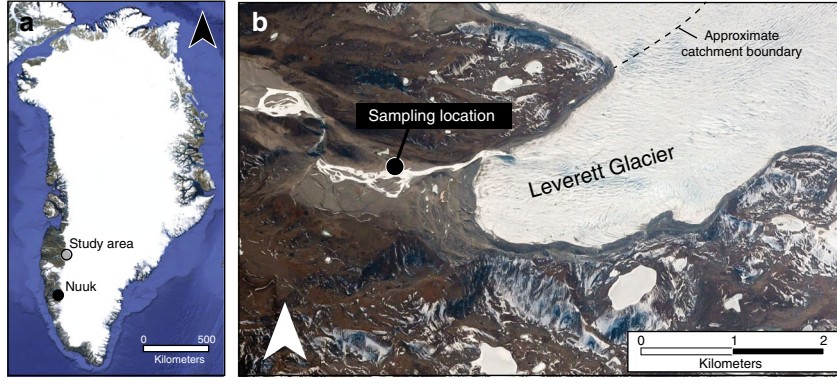

**Fig. 1** Location of Leverett Glacier. **a** Indicates the location of the study area in southwest Greenland. Samples were collected from the proglacial river of Leverett Glacier (**b**) as per Hawkings et al.[22, 44, 64] and Lawson et al.[83]. Image (**a**) of Greenland is from Landsat (US Geological Survey, via Google), and image (**b**) of Leverett Glacier terminus is from DigitalGlobe (via Google)

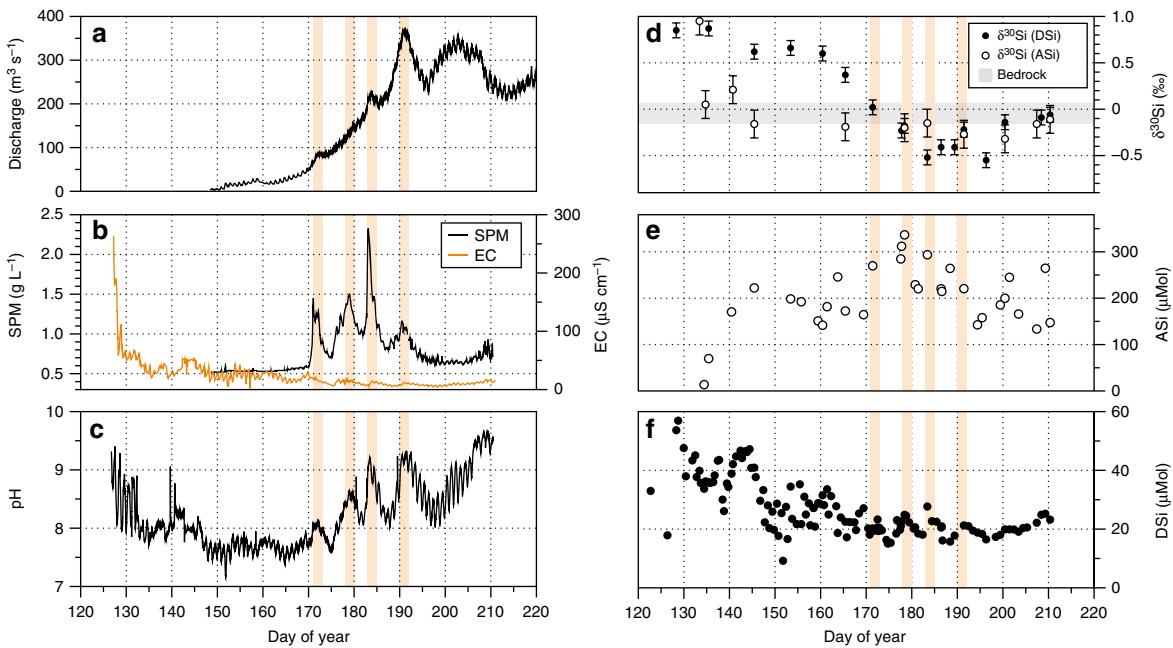

**Fig. 2** Hydrological and geochemical time series from Leverett Glacier proglacial river. **a** Bulk meltwater discharge, **b** suspended particulate matter concentration (SPM) and electrical conductivity (EC), **c** continuous pH time series, **d** Si isotope composition ($\delta^{30}$Si) of dissolved silica (DSi—closed symbols) and reactive amorphous silica (ASi—open symbols) with s.d., alongside suspended sediment and bedrock $\delta^{30}$Si range ($n = 7$) in the shaded grey region, **e** ASi concentrations, and **f** DSi concentrations. The approximate timing of meltwater outburst events is marked with orange shading

**Si isotope three-box model**. We used a modified version of a previously published three-box ocean model[33] with an ensemble of 50 simulations (Supplementary Fig. 2) as a thought experiment to simulate plausible impacts of enhanced meltwater DSi and ASi fluxes from the LGM to the Holocene on the marine Si budget (Figs. 4 and 5; Methods). This model is used as a tool to see how the signal of changing glacial Si fluxes and their associated $\delta^{30}$Si composition would propagate into the ocean in the absence of any changes in marine Si cycling. We estimate a change in the DSi + ASi flux of −39 to +6‰, and a change in $\delta^{30}$Si of the input flux of +0.15 to +0.43‰ from LGM conditions to present day, while MWP1a produces short-term decreases in the $\delta^{30}$Si of total Si input of ~0.1‰ to ~0.2‰ (Supplementary Table 1; Fig. 4). Glacial Si fluxes (and associated changes in nonglacial riverine fluxes) account for between 0.06 and 0.17‰ of the variation in the $\delta^{30}$DSi of the surface boxes and the deep ocean over the past

~21,000 years, with a further excursion of 0.01–0.08‰ during peak meltwater input (MPW1a and MWP1b). The results show relatively large changes in the ocean Si inventory over the deglaciation (Fig. 5), with an increase in the DSi concentration of up to 12.5 µM in the deep ocean at ~10,000 years before present, in response to deglacial meltwater inputs (Fig. 5). As Si input begins to decrease after the deglacial maximum, whole-ocean Si concentrations begin to decrease as well, with the strength of this decrease scaling directly with the LGM–present difference in total Si input flux. Model results further indicate that MWP1a[20] and MWP1b[21] impart a signature on marine DSi and $\delta^{30}$Si (Fig. 5), even though they are relatively short-lived events of a few hundred years[34]. This is especially evident in the larger low-latitude surface box (essentially the surface ocean outside of the Southern Ocean), where MWP1a leads to an excursion in $\delta^{30}$Si of up to −0.08‰.

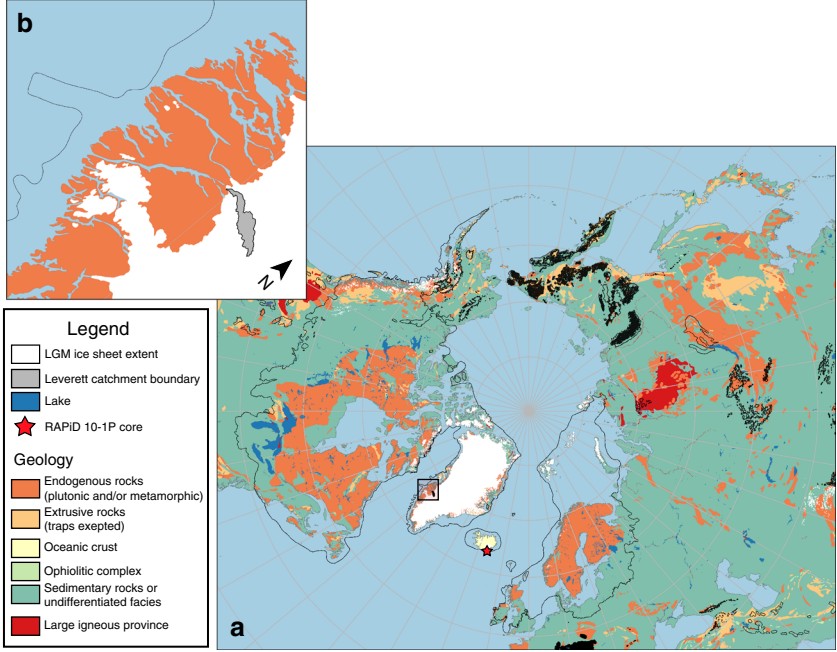

**Fig. 3** Geological map of the Arctic with past ice sheet extent. **a** Arctic Polar Stereographic map with maximum palaeo-ice sheets extent (~21,000 years ago) indicated by a thick black line[17], and insert (**b**) of the Leverett Glacier region (indicated by a red square on the west coast of Greenland in (**a**)) where samples were taken. The estimated Leverett Glacier catchment[84] is shown in **b** by the filled grey area. Reproduced with the permission of OneGeology. Map created using QGIS

**Si isotope record in an Arctic marine sediment core**. A high-resolution record of spicule $\delta^{30}$Si, extracted from a sediment core off southeast Iceland (RAPiD 10-1P; 62.98°N, 17.59°W; 1237 m water depth[35]) was used to investigate possible changes in isotopic inputs in the high-latitudes during a glacial termination, and corroborate box model findings (Fig. 5). The spicule $\delta^{30}$Si record reflects both DSi concentration and $\delta^{30}$DSi of the water in which it was formed[6]. This new record reveals high-frequency variability in spicule $\delta^{30}$Si during ice sheet collapse at ~14,500 and ~11,500 years before present (Fig. 6), concurrent with fluctuations in carbonate stable isotopes[35], assessed from paired planktonic-benthic foraminiferal records. Excursions in the spicule $\delta^{30}$Si of more than −0.5‰ can be observed at both MWP1a and MWP1b.

## Discussion

A striking feature of the isotopic composition of ASi from the GrIS is the ~0.2‰ fractionation from the bedrock signature of 0‰ (0.00 ± 0.07‰; Fig. 2). The consistently lighter $\delta^{30}$ASi signature indicates it is a siliceous precipitate or secondary weathering product[4]. However, it is at the heavier end of previous measurements[4], and we are still uncertain of its origin[14]. Previous studies have suggested ASi forms through mechanochemical action[36,37], dissolution-precipitation[38,39] and as a surface layer left from the preferential leaching of cations during mineral dissolution[40].

The mean $\delta^{30}$DSi is the lightest isotopic composition recorded for riverine waters (glacial and nonglacial). It is lower than the only other measurements of $\delta^{30}$DSi in glacial meltwaters from glaciers in Iceland (0.02 ± 0.18‰)[12,13], and much lower than global rivers (mean = 1.25 ± 0.68‰)[4]. It is also lower than the estimated mean groundwater $\delta^{30}$DSi (0.19 ± 0.86‰)[4], and most similar to measurements of hydrothermal fluids (−0.30 ± 0.15‰)[4]. The anomalously low $\delta^{30}$DSi signal from glacial meltwaters requires either a light $\delta^{30}$DSi source or a heavy $\delta^{30}$DSi sink. The

latter has only been documented with acidic hydrofluoric leaching of basalts[41]; therefore, secondary phase dissolution or rapid leaching of the mineral surface is more likely the source. The glacial meltwaters are significantly undersaturated with respect to ASi (mean $SI_{ASi} = -2.1$)[14] and have high pH (Fig. 2c), so ASi is very likely to dissolve in meltwaters. Further evidence of this is given by the similar discharge-weighted mean for $\delta^{30}$DSi and $\delta^{30}$ASi meltwater composition. It is therefore possible that the lower $\delta^{30}$DSi composition from day ~170 onward is derived from the dissolution of ASi and the partial dissolution of other lighter secondary weathering products (e.g. clays)[42], which may have an even lighter $\delta^{30}$Si signature than ASi[43], explaining the lowest $\delta^{30}$DSi values observed. Dissolution of secondary weathering products is thought to occur in long residence time groundwaters[43], and could reflect drainage of more isolated subglacial water sources further into the glacial catchment as the melt season progresses[44]. It is possible that either glacial chemical weathering preferentially removes $^{28}$Si during initial dissolution of silicate surface layers[5], heavier $^{30}$Si isotopes have either been stripped out during previous chemical weathering (e.g. when glaciers retreated during previous interstadials and interglacials over weathered soils), or heavier isotopes are retained in the mineral weathering crust, to balance the long-term isotopic mass balance of the system.

The time series of $\delta^{30}$DSi also shows a significant temporal shift of >1.3‰ toward lower values from early season low discharge to peak discharge waters, while $\delta^{30}$ASi shows little temporal variation (Fig. 2d; May−July). This likely reflects the weathering environment in which DSi is generated, as described above. The second lowest $\delta^{30}$DSi values were recorded on day 184 (−0.52‰), during a high discharge meltwater outburst event (~230 m³ s⁻¹) characterised by a large spike in SPM concentration, electrical conductivity and pH (Fig. 2—red-shaded region). This indicates that flushing of long-term stored waters at the ice sheet bed is likely to contribute a very low $\delta^{30}$DSi signature.

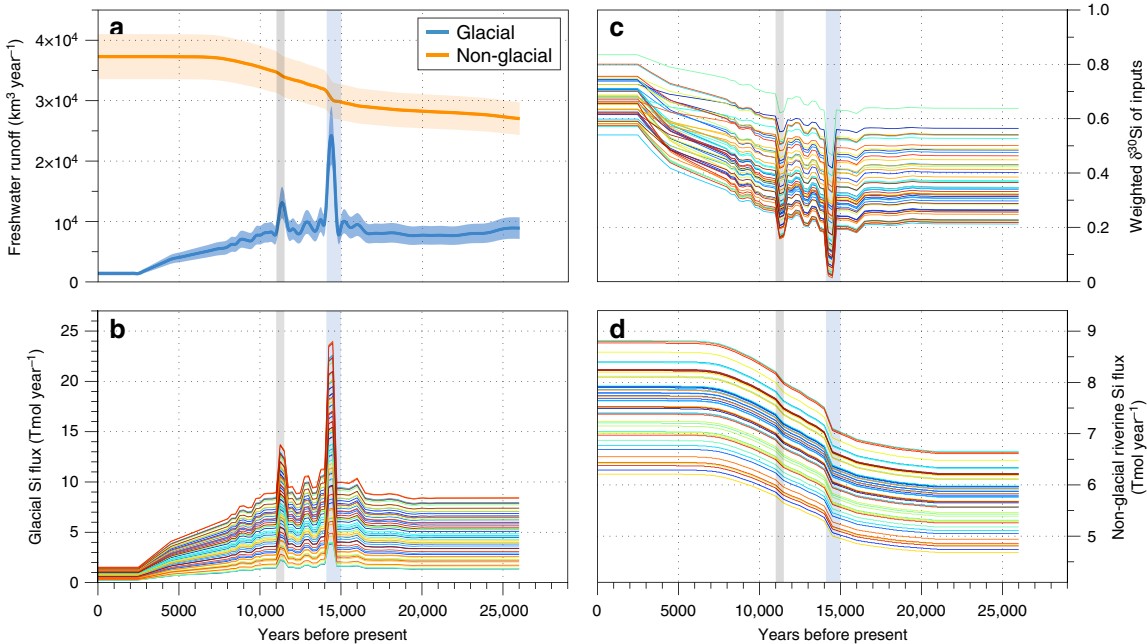

**Fig. 4** Freshwater flux estimates and model input comparison. **a** Freshwater fluxes from glacial and nonglacial sources, with estimated uncertainty given in the orange- and blue-shaded regions, and **b** the glacial DSi + ASi input flux, **c** weighted $\delta^{30}$Si composition of the Si input flux and **d** nonglacial riverine DSi + ASi input flux over each model simulation. Coloured lines correspond to the model simulations in Fig. 5. Additional input data are detailed in Supplementary Table 1. Meltwater Pulse 1a (MWP1a) is highlighted by the shaded blue region, and Meltwater Pulse 1b (MWP1b) by the shaded grey region

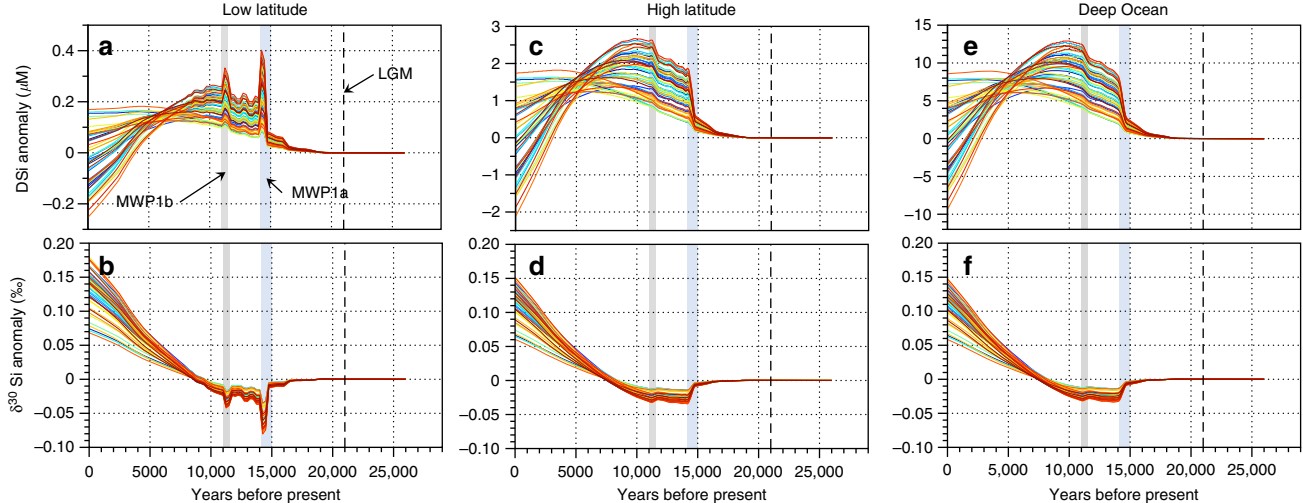

**Fig. 5** Modelled impact of glacial to interglacial ice sheet wastage on the oceanic Si cycle. **a**, **b** Reflect low-latitude surface results for DSi concentrations and $\delta^{30}$DSi anomalies, respectively, **c**, **d** reflect high latitude (i.e. Southern Ocean) DSi concentration and isotopic composition anomalies, respectively and **e**, **f** reflect deep ocean DSi concentration and isotopic composition anomalies, respectively. Results were generated from an ensemble of 50 model simulations (coloured lines) to sample uncertainty in input fluxes and $\delta^{30}$Si composition, and burial and export efficiency (Fig. 4). Input variables for glacial (DSi + ASi flux and respective $\delta^{30}$Si composition), nonglacial (DSi + ASi flux and respective $\delta^{30}$Si composition), export efficiency to depth and burial efficiency in sediments were chosen using a Latin Hypercube. Each simulation was run from a Last Glacial Maximum (LGM) baseline, which was a 100,000-year spin up under LGM conditions (see Methods). Simulations are displayed as anomalies from LGM conditions. The LGM is highlighted by a dashed line, Meltwater Pulse 1a (MWP1a) by the shaded blue region, and Meltwater Pulse 1b (MWP1b) by the shaded grey region. Transient model inputs are displayed in Fig. 4 and Supplementary Table 1

The Si isotopic composition and high DSi + ASi concentration of GrIS meltwaters, combined with previous measurements of Icelandic glacier meltwaters[12], suggest that glacially derived Si is a previously underappreciated source of light Si isotopes to the ocean, either as DSi[24] or dissolvable ASi[14]. We hypothesise that

over periods of time similar to, or longer than, the residence time of silicon in seawater (10–15,000 years)[1,4], changes in glacial land coverage would affect the ocean's Si inventory and isotopic composition, for example over glacial cycles[4]. On shorter (e.g. millennial) temporal and spatial scales, the extremely low $\delta^{30}$Si of

glacial meltwater could influence regional marine silicon cycling and isotopic budgets. The extremely light isotopic signature of GrIS runoff is likely to be broadly representative of PIS during deglaciation. Leverett Glacier is likely to be a crude analogue of ice sheet meltwaters because it is a large glacial catchment (~600 km²), has bedrock geology broadly representative of the shield bedrock that underlies much of the land on which the Eurasian and North American ice sheets sat (Fig. 3), displays characteristic GrIS catchment sediment export and therefore physical erosion dynamic[45], and has a hydrological regime thought to be typical of large outlet glaciers[46–48].

The simulated temporal evolution of glacial and riverine Si inputs to the ocean from the LGM to the present day leads to a switch from glacially dominated Si supply to nonglacial riverine-dominated supply in the present (Fig. 4). All model ensemble members show a broad maximum of Si input during the deglaciation, with sharp peaks during the meltwater pulses (Fig. 4). The $\delta^{30}$Si value of total Si input into the ocean varies over the deglaciation, decreasing sharply during meltwater pulses of isotopically light glacial Si, but generally increasing over the deglaciation as isotopically heavier riverine Si becomes progressively more important than low $\delta^{30}$Si glacial Si input (Fig. 4; Supplementary Table 1).

The three-box model used as a thought experiment to simulate potential oceanic response to changes in glacier silica fluxes (Supplementary Fig. 2) suggests an expanded ocean Si inventory at the LGM and a lower marine $\delta^{30}$Si signature (Fig. 5). This simple experiment includes only variable weathering fluxes in the absence of any changes in marine Si cycling, and as such it is only intended to estimate the magnitude of changes caused by glacial weathering for comparison to available data. The lighter surface $\delta^{30}$Si predicted at the LGM (by 0.06–0.18‰) explains a portion of the glacial to present-day $\delta^{30}$Si increase found in diatomaceous remains, which is usually quoted as 0.5–1.0‰[4,10]. Much of this increase is reliant on ASi dissolution and bioavailability to marine organisms. ASi dissolution is catalyzed by the presence of the alkali metals[49,50], which are found in high dissolved concentrations in seawater. Furthermore, a recent study demonstrates rapid dissolution of ASi from glacial SPM in natural seawater (up to 25% in less than 30 days), at high sediment loading concentrations (1 g L⁻¹ of SPM)[14], corroborating previous evidence of synthetic ASi dissolution in artificial seawater[51]. The isotopic signature from ASi will likely imprint upon ocean waters upon dissolution.

There is an increase in whole-ocean $\delta^{30}$Si between the LGM and present day, as a result of the change in partitioning between isotopically heavy nonglacial river waters and isotopically light glacial meltwaters, in all ensemble members. The change in whole-ocean $\delta^{30}$Si reflects the change in the isotopic composition of the inputs (~0.15 to ~0.45‰), when the model is run to equilibrium (Supplementary Fig. 3). This indicates the modern ocean might still be responding to LGM and deglaciation meltwater inputs, and could continue to do so for at least another ~20,000 years.

The $\delta^{30}$Si value of Si exported from the surface of the high-latitude box (i.e. a simulated opal export flux; Fig. 5) evolves according to the change in whole-ocean $\delta^{30}$Si. This is as expected for the three-box model, in which no change in Si utilisation was simulated in order to isolate the effect of external inputs (i.e. glacial meltwater versus nonglacial meltwater inputs) on the isotopic composition recorded in the marine diatom record. The modelled change in external input (i.e. glacial vs. nonglacial runoff) can explain only around ~0.1‰ of LGM–present change in marine diatom $\delta^{30}$Si records in this box (i.e. up to 20% of the observed change in Southern Ocean diatom core records), which is analogous to the Southern Ocean. There are several possible

candidates to explain the discrepancy between modelled and observed $\delta^{30}$Si. First, an obvious candidate for the observed change is a difference in Si utilisation between the LGM and today[7]. Second, changes in external inputs not modelled in our simulations such as the input of Si through the dissolution of aeolian dust, or change in the nonglacial weathering regime[4], may have a further impact.

The model predicted that whole-ocean DSi concentrations were higher during the deglaciation and likely higher during the LGM than present day (Fig. 5). A larger Si inventory has implications for $CO_2$ drawdown and ecosystem function, via increased diatom productivity, possibly at the expense of other phytoplankton groups[52], as has been observed in Greenlandic fjords[24]. This has important implications for marine biogeochemical cycles, as higher Si input favours the growth of diatoms relative to other phytoplankton groups[53]. This is likely to have an impact on the organic carbon export (due to opal ballasting), surface alkalinity (by changing the proportion of silicifiers to calcifying phytoplankton), and the "silica pump", which controls the ratio of nutrients reaching the deep ocean[52–55]. The model results further indicate that MWP1a and MWP1b[34] impart a signature on marine DSi and $\delta^{30}$Si (Fig. 5), even though they are relatively short-lived events of a few hundred years[34]. MWP1a is especially notable as it leads to whole ocean excursions in lower $\delta^{30}$Si (up to −0.08‰) and elevated DSi concentrations (up to 1.5 µM) in all simulations (Fig. 5—shaded blue). There is likely some influence of iceberg calving in the sea level rise observed during MWP1a. Although inputs of ASi from iceberg rafted debris may be significant[14], this is not currently accounted for in the flux calculations and $\delta^{30}$Si composition of inputs. However, there is a growing consensus that around half of the sea level rise from MWP1a comes from the interior of the Laurentide ice sheet[56], with smaller contributions from Antarctica (likely <2 m[57]), Eurasia (~2.5 m[58]) and Greenland (~0.5 m). Runoff from melting terrestrial ice therefore likely made up the dominant portion of MWP1a freshwater flux and there is little evidence to suggest iceberg calving contributed anywhere near as much to sea level rise during this period. The importance of iceberg Si inputs from large ice calving events (e.g. Heinrich events such as H1) are not included in our model but should be addressed in future research due to the potentially large associated Si fluxes[14].

While the whole ocean excursion during the meltwater pulse events is only of the order of the uncertainty on a $\delta^{30}$Si measurement, we would expect proximal downstream PIS effects to be more pronounced than whole ocean model simulations indicate (as per our sponge spicule record; Fig. 6). Although our model is a relatively crude representation of real-world complexity, it indicates that changes in continental ice sheet coverage and meltwater DSi/ASi input were likely of significant importance in the global Si (and by extension carbon) cycle over these time periods and, by extension, over previous glacial cycles[26].

Data from the sponge spicule record of a marine sediment core proximal to Iceland reveal high-frequency variability in spicule $\delta^{30}$Si during ice sheet collapse, of up to −0.6‰ over ~300 years (Fig. 6). These changes may have been driven by a doubling of DSi concentrations (from approximately 20 to 40–50 µM[6]), variation in seawater $\delta^{30}$DSi at the time of spicule formation, or a combination of both increased DSi concentrations and lower seawater $\delta^{30}$DSi. Such significant and rapid changes would require a local source that is highly enriched in DSi and/or contribute low $\delta^{30}$DSi to ocean surface waters, which our data indicate could be of glacial origin[14,59]. Data from Icelandic glaciers, including a spot measurement from Skeiðarárjökull (which has a catchment area ~1400 km²; $\delta^{30}$DSi of 0.01‰ and DSi concentration of 70.4 µM), corroborates the observation that an enriched and light $\delta^{30}$DSi source from local glacial meltwaters is

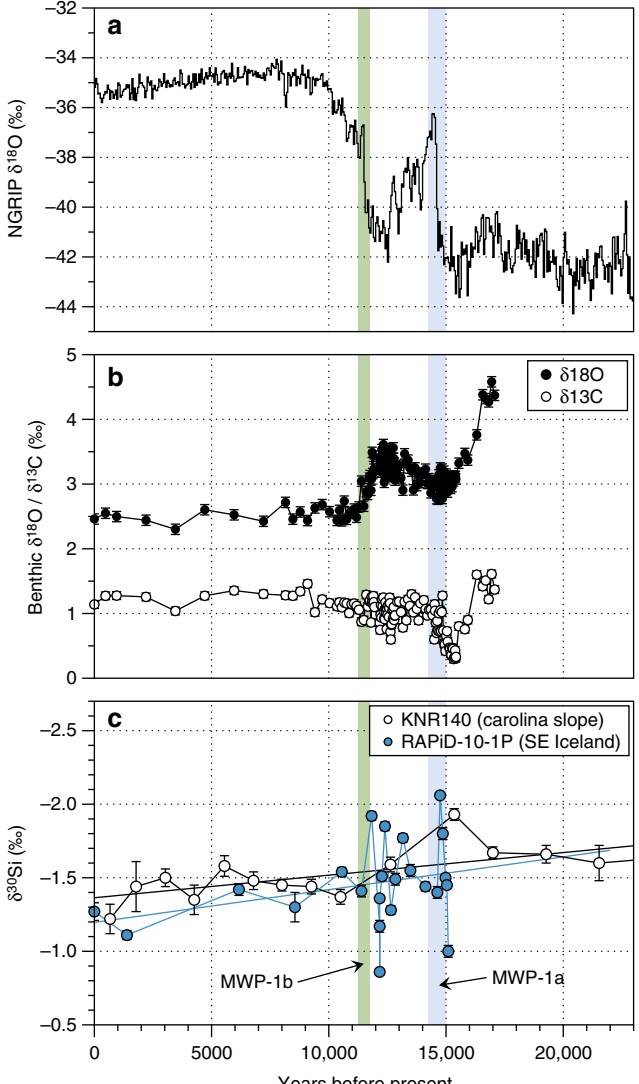

**Fig. 6** Si isotope core records from deep-sea sponge spicules and other palaeo-data. **a** Greenland NGRIP ice core oxygen isotope record[85], **b** benthic $\delta^{18}O$ and $\delta^{13}C$ record from *Cibicidoides* in the RAPiD-10-1P core (SE Iceland)[62], and **c** Si isotope record from sponge spicules for the RAPiD-10-1P core (blue) and the lower resolution KNR140 core for comparison (white; Carolina Slope[8]). The shaded regions indicate approximate timings of Meltwater Pulse event 1a (blue) and Meltwater Pulse event 1b (grey)

likely[12,13]. This interpretation is supported by the observation that the excursions in spicule $\delta^{30}Si$ from our record coincide, within the age model uncertainty, with estimates of flux pulses derived from sea-level changes and collapse of the local Icelandic ice sheet 15,000 to 14,700 years before present[59] (Fig. 6). Further evidence for this comes from the coincidence of the lowest Si isotope values with fluctuations in carbonate stable isotopes[35], assessed from paired planktonic-benthic foraminiferal records[60] (Fig. 6). This signal is consistent with high-frequency switching between the influence of Arctic seawater and glacially derived waters at our study site; the isotopically high and low-$\delta^{30}Si$ water arising from radiocarbon depleted and Si-poor Arctic seawater[61] and comparatively well-ventilated subglacial input, respectively. Although there was likely reduced deep-water ventilation during the early part of the deglacial, this would likely have only influenced deeper waters (>2 km)[62]. Core evidence suggests that

localised polar influence from meltwaters, sea-ice formation and brine rejection would likely have transferred a surface $\delta^{30}Si$ signal to depth via overflow waters[60,62]. These overflow waters originated locally and from the Nordic Seas[62], which would have been heavily influenced by ice melt from northern hemisphere ice sheets, with a corresponding light $\delta^{30}Si$ composition and elevated Si concentrations.

The part played by ice sheets and glaciers in marine nutrient and nutrient isotope cycling is only just starting to be appreciated[1,14,63]. Here we show that these systems act as a significant source of isotopically light Si, either directly via dissolved silica or indirectly as dissolvable amorphous silica attached to SPM. This has the potential to impact the marine Si inventory over a range of different spatial and temporal scales, given the meltwater input from the wastage of large palaeo-ice sheets since the LGM. Our model indicates that the magnitude of meltwater-derived Si inputs is sufficient to drive significant changes in the ocean's Si inventory on glacial/interglacial and deglacial time-scales, thereby modulating the productivity of diatoms relative to other primary producers. Results provide evidence for significant low $\delta^{30}Si$ release during rapid ice sheet wastage in the $\delta^{30}Si$ composition of a high latitude North Atlantic sediment core record, which corroborates the hypothesis that terrestrial ice cover impacted the oceanic Si cycle, derived from modern ice sheet data and modelling results. These findings highlight the important role played by glacial meltwater in the marine Si cycle, aiding in our interpretation of palaeoceanographic proxies and our understanding of past and present carbon cycling. Our data demonstrate the potential for a feedback between PIS growth and decay, increased Si delivery to the ocean and $CO_2$ drawdown via stimulating the productivity of diatoms.

## Materials And Methods

**Hydrological monitoring**. Leverett Glacier runoff was hydrologically gauged from the onset (28 May) to the end of the ablation season (15 September) 2015. Gauging of Leverett Glacier meltwater river has been extensively discussed by others (see, e.g., refs. [22,30,32,48,64,65].). Stage (for conversion to discharge), electrical conductivity and turbidity (a proxy for suspended sediment concentration) were logged every 10 min at a stable bedrock section of the river ~2.2 km downstream of the glacier portal. Permanent (fixed in place) and mobile (re-located to keep pace with river stage) temporary pressure transducers monitored stage, which was converted into discharge using a stage-discharge rating curve of 26 Rhodamine-WT dye dilution injections ($R^2 = 0.81$ for permanent pressure transducer, and 0.84 for the mobile pressure transducer). Discharge was calculated by dividing the amount of dye injected by the area under the return curve. The errors associated with discharge measurements are ±12.1 % following the methods of Bartholomew et al.[32]. Calibrating turbidity (in mV) against 23 manually collected samples (using a USDH48) allowed formation of a continuous suspended sediment record (in g L$^{-1}$). Around 300–500 mL of meltwater was filtered through a pre-weighed 47 mm 0.45 μm cellulose nitrate filter (Whatman®), with the amount of meltwater filtered accurately measured using a measuring cylinder. On return to labs in the UK, filters were oven dried overnight at 40 °C and reweighed to four decimal places. Suspended sediment concentration was plotted against the turbidity at sampling time points to derive a linear relationship ($R^2 = 0.73$). The linear regression between suspended sediment and turbidity was used to derive suspended sediment concentrations at each 10-min interval over the measurement period. Errors associated with suspended sediment measurements are estimated to be ±6%[30].

**Sample collection**. All water samples were collected from the same location ~1 km from Leverett Glacier terminus (Fig. 1) using 1 L HDPE bottles (Nalgene™) and filtered immediately through 0.45 μm cellulose nitrate filters (Whatman®). Samples were stored in clean HDPE 30 ml Nalgene bottles and kept refrigerated until analysis. Cellulose nitrate filters were retained and also stored refrigerated until ASi and bulk SPM analysis.

**Dissolved silica (DSi)**. Dissolved silica (as silicic acid) was determined using LaChat QuickChem® 8500 series 2-flow injection analyser (QuickChem® Method 31-114-27-1-D). The methodological limit of detection was 0.3 μmol, precision was ±1.3% and accuracy was +2.1%, as determined from five replicates of a 250 μg L$^{-1}$ (8.9 μmol) standard prepared by gravimetric dilution from a 1000 mg L$^{-1}$ Si stock (CertiPur®).

**Amorphous silica (ASi)**. Amorphous silica was measured using the weak alkaline extraction method of DeMaster[66], used to determine biogenic opal and, increasingly, inorganic amorphous silica in terrestrial soils and sediments[67,68]. The DeMaster[66] method uses 0.1 M sodium carbonate ($Na_2CO_3$) solution, a weak base, which maximises the dissolution of amorphous Si with minimal impact on more refractory crystalline material. Approximately 30 mg of sample (accurately weighed) was placed in a clean 60 mL HDPE bottle (Nalgene™) with 50 mL of 85 °C 0.1 M $Na_2CO_3$ solution. Bottles were placed in a hot water bath at 85 °C for the duration of the extraction. Aliquots of 1 mL were taken at 2, 3 and 5 h and stored refrigerated in a new, clean 2 mL microcentrifuge tube (polypropelene). Samples were measured using the same method as DSi (above) within 24 h immediately after dilution of 0.5 mL sample in 4.5 mL of 0.021 M HCl. At least two blanks were processed alongside samples. Precision was ±2.9% and accuracy was +0.4%, as determined from five replicates of a 250 µg L$^{-1}$ (8.9 µmol) standard prepared by gravimetric dilution from a 1000 mg L$^{-1}$ Si stock (CertiPur®). Amorphous silica was determined by using the intercept of the regression line. This method was also compared to the method of extracting total reactive silica for $\delta^{30}$ASi using 0.2 M sodium hydroxide (NaOH), to ensure consistency between % ASi results and silicon isotope composition (Supplementary Fig. 4). We are confident this method extracted mostly ASi as % extractable Si was lower or similar to the well-tested 0.1 M $Na_2CO_3$ method.

**Silicon isotope analysis**. To determine $\delta^{30}$ASi, approximately 10–30 mg of sample was air-dried in a laminar flow hood and accurately weighed into Teflon vials (Savillex). To this, 1 mL 0.2 M NaOH was added per mg of sediment. Samples were heated at 100 °C for 30 min to extract reactive silica (assumed to be mostly ASi; Supplementary Fig. 4). This method was compared to the method of extracting ASi using 0.1 M $Na_2CO_3$, to ensure consistency between %ASi results and silicon isotope composition (Supplementary Fig. 4). We are confident this method extracted mostly ASi as % extractable Si was lower or similar to the well-tested $Na_2CO_3$ method.

Samples were then acidified with 8 N nitric acid (30 µL per 1 mL of solution), diluted 1 in 5, and then centrifuged for 5 min at 4000 rpm before being filtered through a 0.22 µm PES syringe filter (Pall Acrodiscs). A threshold minimum value for %ASi was used for samples (0.01% ASi), as samples with very low %ASi may have resulted in more refractory material also being extracted, skewing the $\delta^{30}$Si record.

Bedrock ($n = 3$) and bulk SPM ($n = 4$) $\delta^{30}$Si were determined after alkaline fusion, adapted from the method of Georg et al.[69]. Bedrock samples (unsorted, coarse proglacial debris collected at the front of Leverett Glacier) were initially crushed using a sledgehammer on a metal plate (within multiple thick polyethylene bags), then ground for 1 min in a Fritsch Planetary Mono Mill Pulverisette 6 at 500 rpm, following the methods of Telling et al.[70]. Approximately 15 mg of bulk rock sample powder or suspended particulate material was subsequently accurately weighed into a silver crucible with ~200 mg of NaOH pellets (analytical grade). Crucibles were then placed in a muffle furnace, heated to 730 °C for 10 min to fuse, and allowed to cool for 10 min. Samples were added to 30 mL of deionised water (18.2 MΩ cm$^{-1}$ Milli-Q Millipore), left overnight and then sonicated for 10 min to aid final dissolution. Samples were acidified and diluted with Milli-Q water using a ratio of 2.1 mL 8 N $HNO_3$ per 500 mL water, before analysis as below.

Dissolved samples were preconcentrated in Teflon vials by evaporating on a hotplate at 90 °C until approximately 2 ml of sample remained. All samples (equivalent 7.2 µg Si) were purified using precleaned BioRad exchange resin AG50W-X12 (in H$^+$ form) columns and eluted with MQ water[3], before being spiked with a Mg solution (Inorganic Ventures). Freshwater samples and their bracketing standards were additionally spiked with 50 µL 0.01 M sulphuric acid (Romil-UpA) to reduce mass bias resulting from high anion loading[71]. Silicon isotope composition ($^{28}$Si, $^{29}$Si, $^{30}$Si) was determined by mass spectrometry using a Thermo Scientific™ Neptune Plus™ High Resolution MC-ICP-MS in the Bristol Isotope Group laboratories at the University of Bristol (Supplementary Table 3). Machine blanks were <1% of the signal on $^{28}$Si, and procedural blanks were below the limit of detection. A standard-sample-standard bracketing procedure with Mg-doping was used to correct for mass bias[72]. International reference NBS-28 was used as the bracketing standard and sample results were calculated using the $\delta^{30}$Si notation for deviations of $^{30}$Si/$^{28}$Si from this bracketing standard (Eq. (1)).

$$\delta^{30}Si = \left[ \frac{(^{30}Si/^{28}Si)_{sample} - (^{30}Si/^{28}Si)_{NBS28}}{(^{30}Si/^{28}Si)_{NBS28}} \right] \times 1000. \quad (1)$$

$\delta^{30}$DSi internal precision was typically ±0.10‰ (2σ) for $\delta^{30}$Si and ±0.05‰ (2σ) for $\delta^{29}$Si. The long-term reproducibility was determined by analysis of two international reference standards, characterised by a number of research groups. The mean for diatomite was +1.26‰ ± 0.11‰ (2σ) for 27 measurements[73] and the average for LMG08 (sponge) was −3.33‰ ± 0.15‰ (2σ) for 53 measurements[74]. External reproducibility of freshwater $\delta^{30}$Si was assessed using a lake water standard (RMR4) from the NERC Isotope Geosciences Laboratory UK (NIGL), which had mean $\delta^{29}$Si and $\delta^{30}$Si values of +0.46‰ ± 0.02‰ (2σ) and +0.91‰ ± 0.03‰ (2σ) respectively ($n = 3$) in good agreement with previous measurements from NIGL[75]. A three-isotope plot of all the samples measured during the study

can be plotted along a straight line with a gradient of 0.523 ± 0.025 (0.526 showing mass-dependent fractionation; Supplementary Fig. 5).

**Flux estimates for the silicon isotope three-box model**. Glacial meltwater flux from the palaeo ice sheets into the oceans is calculated using the ICE-6G_C (VM5a) reconstructed ice mass loss leading to sea level rise from the LGM to present day (Supplementary Fig. 6)[34,76]. The timestep of the reconstruction is 500 years. Thus, although shorter meltwater pulse events (such as MWP1a and MWP1b) are not precisely resolved in time, their meltwater contribution to the global oceans is captured in the longer term. In addition to this deglaciation meltwater, we include a flux from modelled precipitation minus evaporation (i.e. meltwater runoff that is balanced by snow and ice accumulation) over the palaeo ice sheets[77], which was calculated from the best 30 members of an ensemble of simulations validated against both the preindustrial and LGM climate. At the LGM, this yields an ice sheet runoff estimate of ~7700 ± 770 km$^3$ year$^{-1}$, peaking at 21,600 ± 2160 km$^3$ year$^{-1}$ during the period encompassing Meltwater Pulse 1a (MWP1a), and falling to ~1400 ± 140 km$^3$ year$^{-1}$ at present[78]. These values give a reasonable first-order approximation of changes in glacial runoff over the last 21,000 years.

Riverine runoff input fluxes are calculated using two end members: modern-day runoff (37,288 ± 1846 km$^3$ year$^{-1}$)[79] and the percentage difference in precipitation −evaporation simulated over non-ice covered land area for the modern day compared to LGM using the same 30 climate model ensemble as above (−24.6%; 28,115 ± 1391 km$^3$ year$^{-1}$)[77]. Changes in runoff at each time point during the deglaciation are approximated by scaling to the percentage land ice cover[34] from these two end members.

Estimates for other major silica (DSi + ASi) input fluxes (groundwater, aeolian dust, hydrothermal and sea floor weathering) are taken from Tréguer et al.[1] and Frings et al.[4]. Aeolian dust fluxes are known to change significantly from the glaciation to present day[80] but are kept constant in our simulations to allow for evaluation of glacier meltwater effect only. Riverine DSi and ASi concentrations and associated $\delta^{30}$Si composition are taken from Durr et al.[81] (DSi), Conley[82] (ASi) and Frings et al.[4] ($\delta^{30}$DSi and $\delta^{30}$ASi composition) to calculate fluxes and $\delta^{30}$Si of riverine inputs at each model time step. Glacial meltwater DSi and ASi fluxes and $\delta^{30}$Si composition are taken from samples measured at Leverett Glacier in this paper (Supplementary Table 1). We estimate a change in the DSi + ASi flux of −11 (−39 to +6)%, and a change in weighted $\delta^{30}$Si of the input flux of +0.33 (+0.23 to +0.47)‰ from LGM conditions to present day, using these mass balance calculations (with minimum and maximum values; Supplementary Table 1; Fig. 4).

**Silicon isotope three-box model framework**. We adapt the three-box ocean model of Sarmiento and Toggweiler[33] to simulate the deglacial oceanic cycle of Si and its isotopes in an open-system context, i.e. with Si inputs into and outputs from the ocean. The inputs to the ocean are computed as described above (note Heinrich event H1 is not included in our model due to the uncertainties in associated freshwater fluxes), while outputs are parameterised as a temporally constant fraction of export that is lost from the ocean by burial in sediment (see *Model ensemble* below). The model was run at a 1-year time step with a time-transient Si input (magnitude and weighted $\delta^{30}$Si of the input flux) dependent on the balance between ice sheet meltwater flux and riverine flux over the last 21,000 years (from LGM to present day; as below; e.g. Supplementary Tables 1, 2).

The three-box model of Sarmiento and Toggweiler[33] splits the ocean into a deep-ocean box (96.8% of ocean volume) and two surface-ocean boxes, one representing the low latitudes (2.2% of ocean volume, 85% of the ocean surface) and one representing the high latitudes (1% of ocean volume, 15% of the ocean surface). These boxes are connected by a simplified representation of the ocean circulation as represented in Supplementary Fig. 2.

The model's Si cycle is driven by Si uptake in the two surface boxes, which is parameterised as a first-order function of Si concentration, with the rate constants $k_l$ and $k_h$ (see Supplementary Table 1). This uptake drives an export of Si into the deep ocean box. For a given steady-state Si concentration in the surface ocean boxes, a version of the model in which all Si taken up is exported to the deep ocean produces identical results (in terms of isotopic and concentration response) to a version in which 50% of the Si taken up is redissolved in the surface boxes (following e.g. Tréguer and De La Rocha[1]). A small fraction $f_b$ of this export flux does not dissolve within the deep ocean but is lost to burial in sediment, representing the output term that, in equilibrium, balances the input of Si from external sources.

In addition to the glacial and riverine fluxes of Si to the ocean discussed above, other external sources of Si are also included (detailed in the section Flux estimates for silicon isotope three-box model), and are assumed to be constant over the deglaciation in order to isolate the effect of changes in glacial/riverine Si input on the oceanic Si system (Supplementary Table 2). These temporally constant fluxes are simulated following Frings et al.[4], and include input of Si from aeolian deposition, groundwater discharge, hydrothermalism and seafloor weathering, each of which contributes Si to different boxes of the model (see Supplementary Fig. 2).

Si isotopes are handled in the model by carrying a tracer of $^{30}$Si. The only process that produces isotope fractionation in the model is the uptake of Si in the two surface boxes; this fractionation is simulated by scaling the rate constant of $^{30}$Si uptake by the fractionation factor $\alpha = 0.9989$ (i.e. an isotope effect of −1.1‰)

relative to that for the Si tracer. No isotope fractionation during dissolution is modelled. The isotope composition of all input fluxes to the ocean (including the temporally variable glacial and riverine fluxes) is simulated as a temporally constant value (see Supplementary Table 2) based on Frings et al.[4]. We carry out an ensemble of 50 simulations in which a Latin Hypercube sampling method is used to choose a range of possible temporal evolutions of Si input fluxes from glacial and riverine fluxes, with the ranges derived as described above. With one exception, all parameters of the ocean-interior Si cycle are left unchanged in the ensemble. The uptake rate constants $k_l$ and $k_h$ are explicitly left unchanged so as to avoid any changes in the relative utilisation of Si between simulations, since we wish to quantify the degree to which the $\delta^{30}$Si of exported particulate Si may change over the deglaciation *without* any change in Si utilisation. The one uncertain parameter of the oceanic Si cycle that does change between simulations is the burial fraction $f_b$. We constrain the range of possible values that $f_b$ may plausibly take in the context of this model by conducting a sensitivity test using a Latin Hypercube sampling approach: we run a 150-member model ensemble in which $f_b$ is varied concurrently with a range of Si input fluxes corresponding to the uncertainties on the modern Si flux to the ocean (riverine fluxes from Frings et al.[4]; glacial fluxes extrapolated from this study as in the main simulations). The resulting dependency of the whole-ocean mean Si concentration on $f_b$ (Supplementary Fig. 7) is used to determine an uncertainty range for $f_b$. As can be seen, the modern mean-ocean Si concentration of ~92 μM [86] is reproduced for values of $f_b$ between 0.056 and 0.073, and we thus apply this range to the model ensemble.

For each member of the ensemble, the model was spun up with the ensemble member's specific values of $f_b$ and LGM input fluxes for 100,000 years, followed by a 21,000-year simulation of the deglaciation. These results are presented in Fig. 5. Continuation of the simulations for a further 79,000 years (i.e. a total of 100,000-year post-spin up) allows us to assess the long-term response of the model. These results are presented in Supplementary Fig. 3.

**Data availability**. The data used in this article are available from the corresponding author (jon.hawkings@bristol.ac.uk) on request.

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

## Acknowledgements

We thank all of those who assisted with fieldwork at LG, Dr. Fotis Sgouridis and Mr. James Williams in LOWTEX laboratories at the University of Bristol, Dr. Chris Coath in Bristol Isotope Group laboratories at the University of Bristol, Prof. Nicholas McCave and Dr. David Thornalley who provided sediment core samples, and Dr. Lauren Gregoire for assistance in preparing the ice mass loss data. We are grateful to Prof. Rob Raiswell and three anonymous reviewers for their constructive comments on the manuscript. This research is part of the UK NERC-funded DELVE (NERC grant NE/I008845/1) and a Leverhulme Trust Research Grant (RPG-2016-439) to J.L.W., and the ICYLAB grant (ERC grant ERC-2015-Stg—678371_ICY-LAB) to K.R.H. The Lever-hulme Trust, via a Leverhulme research fellowship to J.L.W., the Royal Society, via a university research fellowship (UF120084) to K.R.H., and the Czech Science Foundation (GACR), via a junior grant (15-17346Y) to M.S., provided additional support. G.f.d.S. was supported by the European Union's Horizon 2020 research and innovation pro-gramme under Marie Skłodowska-Curie (grant agreement #708407), R.I. was funded by an NERC Independent Research Fellowship (NE/K008536/1), and T.J.K. was supported by Charles University Research Centre (programme no. 204069).

## Author contributions

All authors made a significant contribution to the research presented here. J.L.W., K.R.H., M.T. and J.R.H. conceived the project. J.R.H., J.E.H., T.J.K., M.S., G.L.-G., A.B., E. B. and A.T. collected the field data. J.E.H., K.R.H. and J.R.H. undertook lab analysis. J.R. H., J.E.H., K.R.H. and J.L.W. wrote the manuscript. K.R.H., J.P. and M.T. provided significant help and invaluable advice in lab analysis. G.f.d.S., M.P.H. and R.I. aided in model development, data input, set up and output interpretation.

## Additional information

**Competing interests:** The authors declare no competing interests.

