## [Peer Review File · Nature Communications]

Reviewers' comments:

Reviewer #1 (Remarks to the Author):

Review of NCOMMS-17-14828

Gregory de Souza

In their manuscript, Dr. Hawkings and colleagues present Si concentration and isotopic data from runoff in the catchment of the Leverett glacier in Greenland. Their data demonstrate that this glacier's runoff bears the isotopically lightest dissolved Si thus far measured in surface waters (the only isotopically lighter waters being groundwater and submarine hydrothermal fluids), as well as being enriched in isotopically light amorphous silica, which the first author has previously shown to be labile in seawater (Hawkings et al., 2017). Based on these results, the authors undertake two modelling studies to assess the influence of varying input of isotopically light glacially-derived Si during the last deglaciation on (a) the interpretation of diatom $\delta^{30}\text{Si}$ records from marine sediment cores and (b) the isotopic composition of dissolved Si in the global ocean. Their modelling results imply that glacial-interglacial changes in marine diatom $\delta^{30}\text{Si}$ may not result from changes in Si utilisation, as currently thought, but rather reflect changes in the partitioning of Si supply between glacier-fed and non-glacier-fed rivers via their influence on whole-ocean $\delta^{30}\text{Si}$. At a finer temporal scale, the authors also argue that two large meltwater pulses during the deglaciation may have contributed significant amounts of isotopically light Si to the North Atlantic Ocean. This argument is supported using data that the authors present from a marine sponge spicule $\delta^{30}\text{Si}$ record from the subpolar North Atlantic.

Ocean biogeochemistry change is a key mechanism for shifts in the exogenic carbon cycle that accompanied and exacerbated glacial-interglacial global change. Because the oceanic residence time of Si is similar to the time scale of the deglaciation (10-15kyr), understanding open-system dynamics such as the glacial input studied by the authors is key to the correct interpretation of $\delta^{30}\text{Si}$ records from biogenic silica over this time period, and the author's focus is thus of great relevance to the palaeoceanographic community interested in glacial-interglacial changes in oceanic nutrient cycling. Obviously, their extrapolation from one glacial catchment to the entire glaciated northern landmass is a large leap of faith, but as a thought experiment at least, their application of these early data is valuable. Recent research showing the importance of glacial input for a number of nutrient elements in the ocean makes this work timely as well.

However, I cannot recommend the publication of this paper in its current form. This is mainly because I find the modelling tools that the authors have used to be inappropriate to the questions at hand, in one case fatally so. To a lesser degree, it is because I find one of the models insufficiently documented, and because I have questions regarding the interpretation of the marine sponge spicule $\delta^{30}\text{Si}$ record (although the authors may be able to easily answer these). I discuss my main objections below. Dealing with these would, in my opinion, make this paper a much stronger contribution that could be considered for *Nature Communications*.

Major comments

1. Reconstruction of LGM/present Si utilisation

The manuscript's main claim that "diatom Si [uptake] during the LGM was likely similar to present day" (L10-11) is based on the model of Opfergelt et al. (2013), presented on L105-114 and in the Supplementary Information. Unfortunately, the authors of that study made some exceedingly poor assumptions in formulating their model, assumptions that the present authors have naturally taken over in their entirety. As a consequence, I cannot consider these results to be robust, or even meaningful, as I attempt to explain below.

The Opfergelt et al. (2013) model essentially consists of two equations derived from Rayleigh distillation (Strutt, 1902); one for the substrate (i.e. dissolved Si in the ocean) and the other for the product (i.e. diatom silica). It is worth noting here that the equation Opfergelt et al. (2013) use for diatom silica is the equation for the so-called “instantaneous product” in Rayleigh jargon – i.e. the infinitesimally small fraction of product that is added to the ever-accumulating pool of biogenic silica (“accumulated product”) during any one infinitesimally small time-step. This is the first flaw of this model, since any biogenic silica exported out of the ocean’s mixed layer at timescales of seasons or blooms (and certainly any biogenic silica that accumulates in sediments on the ocean floor) is much more likely to be represented by “accumulated product”, the Rayleigh isotopic dynamics of which are entirely different from those of the instantaneous product. Thus, I fundamentally question the aptness of comparing the results of the Opfergelt et al. (2013) model to marine diatom $\delta^{30}\text{Si}$ records. However, this is not my main objection.

The second, and fatal, problem with the Opfergelt et al. (2013) model is the way in which it deals with the isotopic composition of inputs into the ocean. In the model, the isotopic composition of these inputs *directly* determine the isotope composition of the source ($\delta^{30}\text{Si}_{\text{input}}$) in Eqn. 1 of the Supplementary Information, i.e. the isotopic composition of the initial substrate before any Si uptake by diatom growth. This formulation is problematic in two ways:

- (a) it wrongly assumes that the isotopic composition of the substrate from which diatoms grow (i.e. oceanic mixed-layer DSi) is identical to that of the mean external input to the ocean, which is patently not so (since all oceanic DSi is isotopically heavier than the mean input; e.g. De La Rocha et al., 2000; Frings et al., 2016). This assumption thus sets the starting point of Rayleigh fractionation at an unrealistically low $\delta^{30}\text{Si}$ value, such that any estimates of Si utilisation are certain to be inaccurate.
- (b) more importantly, it assumes that a glacial-interglacial *change* in the isotopic composition of the inputs translates directly into a change in the isotopic composition of the substrate from which diatoms grow. As the authors’ own results from the two-box model show, this is not the case: in their results, the ensemble-mean LGM-to-present shift in the $\delta^{30}\text{Si}$ value of input is $\sim 0.5\text{‰}$ (Fig. 5c), but this translates into only a $\sim 0.25\text{‰}$ ensemble-mean shift in deep-ocean $\delta^{30}\text{Si}$ over this timescale (Fig. 6d), due to the residence time of Si. This assumption thus exaggerates the influence of changes in input $\delta^{30}\text{Si}$ on reconstructions of Si utilisation.

In summary: firstly, the approach first undertaken by Opfergelt et al. (2013) makes an erroneous comparison of sedimentary diatom $\delta^{30}\text{Si}$ values to the “instantaneous product” of the Rayleigh distillation equations, and, secondly, it incorrectly ignores the isotope systematics of the ocean system and the dynamics of the oceanic Si inventory over the deglaciation. Together, these flaws mean that this model simply cannot be meaningfully used to calculate differences in Si utilisation between the LGM and today, and thus that the authors’ claims of unchanged Si utilisation between the LGM and today (L197-205) cannot be considered to have a firm basis. Any model that attempts to understand the influence of transient input change on marine $\delta^{30}\text{Si}$ cannot get around modelling the marine Si inventory as well.

2. Modelling deglacial ocean change

2a. Choice of two-box model: In their two-box modelling study, the authors *do* explicitly model the marine Si inventory and its change over the deglaciation. They have also put a great deal of effort into compiling modelling results and literature data in order to produce a well-defined time history of inputs

into the ocean over the deglaciation (I am not qualified to judge how plausible these are, but the authors seem to have done a thorough job).

Given this considerable effort, I am surprised that the authors have chosen to stick to a two-box model of the ocean. I'm sure I don't need to tell the authors that ever since 1984 (Knox and McElroy, 1984; Sarmiento and Toggweiler, 1984), it has been known that representation of glacial-interglacial ocean biogeochemistry requires consideration of the polar oceans, e.g. through use of a three-box model. The polar oceans are key in controlling the marine cycle of Si and its isotopes (Sarmiento et al., 2007; Brzezinski and Jones, 2015), and it is in the polar Southern Ocean that the clearest glacial-interglacial changes in marine diatom $\delta^{30}\text{Si}$ records are seen. Given the difference in surface Si concentrations between low- and high-latitudes in the modern ocean (and corresponding differences in the extent to which Si isotopes are fractionated), I would expect the deglacial isotopic response of a three-box model to be appreciably different to that of a two-box model. Whilst I appreciate the need for model simplicity over long integration times, models need to be not only as simple as possible, but also as complex as necessary for the task at hand, and I believe a two-box model of the oceanic Si cycle does not fulfil this second requirement. And the additional computational cost of a three-box model would be easily bearable, in my opinion.

I would like to motivate the authors to make this change by saying that, apart from more correctly representing the coarsest-scale behaviour of the ocean system, using a three-box model would allow them to:

- (a) compare the *timescale* of simulated $\delta^{30}\text{Si}$ change with observational records from polar marine sediments, potentially allowing deconvolution of trends associated with whole-ocean change from changes relating to biogeochemical cycling/circulation;
- (b) explicitly model the propagation of a pulse of isotopically light meltwater-derived Si from the North Atlantic to the global ocean;
- (c) compute more meaningful values for simulated surface ocean DSi change than the global-average number they report on L126.

A quantitative comparison between observational records and simulated polar ocean change, with special consideration of the timescale of this change, would be a real step forward in understanding the influence of external inputs on marine $\delta^{30}\text{Si}$ records. In contrast, the authors' current approach limits them to showing that the modelled changes in input cause a change in mean-ocean $\delta^{30}\text{Si}$ of $\sim 0.25\%$ (ensemble mean) over the deglaciation. This is essentially the same result arrived at by Frings et al. (2016) using a similar two-box model, and thus not really novel.

2b. Model documentation: Another issue I have with the two-box model is that there is insufficient detail provided regarding how key behaviours are formulated; without these it remains entirely opaque to the reader. Given the variety of silica production behaviours modelled by De La Rocha and Bickle (2005), some of them rather *ad hoc*, it is not sufficient to simply refer to this paper for documentation. Some key questions I have are:

- How is the uptake of Si in the surface ocean parameterised? Is it a function (linear or non-linear) of surface ocean Si concentration?
- Does whole-ocean volume change with the large prescribed changes in freshwater flux over the deglaciation?

- Following on the two questions above, how does the residence time of Si change across the deglaciation? Does burial respond linearly to changes in input? It might be helpful to show a time-series of Si residence time, at least in the SI.
- Do the authors follow De La Rocha and Bickle's (2005) method of prescribing Rayleigh systematics for Si uptake? If so, can they justify why they do so? Given that the authors simulate two isotopes of Si in their model (as I gather from the SI), I see no reason to do this: a simple parameterisation of kinetic isotope fractionation (i.e. a difference in the rate constants for the two isotope tracers simulated) would suffice, and isotope dynamics would emerge from the model, rather than needing to be prescribed.

It is vital for the reader to know what parameterisations were chosen for uptake and dissolution/burial, since these factors will be a key determinant of how the model's whole-ocean Si inventory responds to changing inputs, as well as what might cause the higher DSi concentrations that the model simulates for the LGM ocean. Furthermore, the isotope dynamics of the surface ocean are crucial in determining the ocean's *isotopic* response. This information should be supplied in the Supplementary Information.

3. Interpretation of sponge $\delta^{30}\text{Si}$ record

Having shown in their two-box model that MWP1a and 1b may have induced short-term changes in ocean $\delta^{30}\text{Si}$, the authors look for evidence of such change in a sediment core from south of Iceland (a setting I would call subpolar North Atlantic, not "Arctic"; L130), from which the authors present a record of sponge-spicule $\delta^{30}\text{Si}$. The authors argue that the high-frequency variability observed in sponge spicule $\delta^{30}\text{Si}$ at this site around the time of MWPs 1a and 1b must reflect changes in the concentration and/or isotopic composition of regional Si input, thereby reflecting regional influence of glacial input.

I have one main question regarding this sponge record, which shows remarkable variability. Could it possibly be that this variability is simply the result of the much higher resolution of this record relative to most sponge $\delta^{30}\text{Si}$ records, rather than actually reflecting greater variability at this site? Whilst I am sure that the authors have taken care to convince themselves that the data represent a real environmental signal, I think the reader needs to be convinced of this too. The Methods and SI provide no information on sampling, $\delta^{30}\text{Si}$ sub-sampling and sample preparation, so it is not possible to determine if, for instance, sample size is comparable between the high- and low-resolution records presented here. It would also support the authors' interpretation if high-resolution records from other locations (or over other time periods) do not show such variability. Unfortunately the only other high-resolution sponge $\delta^{30}\text{Si}$ record I know of is also from just south of Iceland (RAPiD 15-4P; Hendry et al., 2016) and over the same time period, but perhaps the authors know of other data?

A second point that I think the authors should expand at least briefly on is the fact that their sediment core is from a water depth of ~1250m, whilst glacial runoff obviously enters the ocean at its surface. Whilst the North Atlantic is a region of deep water formation today, my understanding is that northern ventilation was shallower during the first half of the deglaciation (and that meltwater pulses are associated with reduced ventilation). I think it would be helpful for the authors to comment at least briefly on the evidence for northern ventilation to the depths of their sediment core location, since this would significantly strengthen their argument.

Finally, comparing the Carolina Slope data shown in Fig. 7 with the data from this core published originally in Hendry et al. (2016), the data points appear to be shifted by a few kyrs. If the core's age model has been revised, this should be mentioned somewhere; if not, it would appear that there is an error in Fig. 7.

Minor comments

Regarding the calculation of past Si utilisation, I would like to point out that the model the authors use has only ever been applied by Opfergelt et al. (2013), and not by De La Rocha et al. (1998) as the authors suggest on L110 and L303. Opfergelt et al. (2013) simply applied their model to data from De La Rocha et al. (1998), and arrived at different values for input to the global ocean based on two differing ocean Si budgets (see their Table 2). Note that De La Rocha et al. (1998), in order to produce a quantitative estimate of utilisation from their diatom $\delta^{30}\text{Si}$ data, assumed a source $\delta^{30}\text{Si}$ value of +1.6‰ (see their last paragraph) as an estimate of the isotopic composition of marine DSi being supplied to the euphotic zone from below.

On L121, the authors state that the LGM-to-present change in the weighted $\delta^{30}\text{Si}$ of the input flux is +0.33‰, but the ensemble mean in Fig. 5c shows a change of around +0.48‰. Is there an error somewhere, or am I misunderstanding the authors?

Fig. 6: from the figures it would appear that simulations are represented as anomalies from *LGM* conditions, and not from present-day conditions as stated in the caption (L643).

Refs not cited in the MS

- Brzezinski, M.A., Jones, J.L., 2015. Coupling of the distribution of silicon isotopes to the meridional overturning circulation of the North Atlantic Ocean. *Deep Sea Res. II* 116, 79-88.
- De La Rocha, C.L., Brzezinski, M.A., DeNiro, M.J., 2000. A first look at the distribution of the stable isotopes of silicon in natural waters. *Geochim. Cosmochim. Acta* 64, 2467-2477.
- Knox, F., McElroy, M. B., 1984. Changes in atmospheric CO₂: influence of the marine biota at high latitude. *J. Geophys. Res.* 89, 4629-4637.
- Sarmiento, J.L., Toggweiler, J.R., 1984. A new model for the role of the oceans in determining atmospheric pCO₂. *Nature* 308, 621-624.
- Sarmiento, J.L., Simeon, J., Gnanadesikan, A., Gruber, N., Key, R.M., Schlitzer, R., 2007. Deep ocean biogeochemistry of silicic acid and nitrate. *Glob. Biogeochem. Cyc.* 21, doi:10.1029/2006GB002720.
- Strutt, J. W. (Lord Rayleigh), 1902. On the distillation of binary mixtures. *Phil. Mag.* 4, 521-537.

Reviewer #2 (Remarks to the Author):

This study uses Si concentrations and Si isotope compositions from a glacier in west Greenland and a Si isotope record of sponge spicules in a marine sediment core off southern Iceland to model input Si flux and $\delta^{30}\text{Si}$ composition during the late Wisconsin and Holocene. The manuscript is interesting in the sense that it expands the previous work by De La Rocha, Treguer, Frings, Opfergelt, Georg and their co-workers, by focusing on the glacial Si component to the ocean Si budget. While this is a valid and praiseworthy study objective, I struggle to understand whether the findings of this study actually improve our current knowledge about the history of the global Si cycle significantly.

The premise of the study is that "riverine silicon (Si) flux and isotope composition ($\delta^{30}\text{Si}$) are thought to be uniform in time and unaffected by the expansion and retreat of large ice sheets during glacial-interglacial cycles" (lines 2-4). I cannot recognize that this is the general understanding within the research community, and I am surprised that the authors believed so before this study was initiated in 2015. In fact, the premise contradicts the general understanding that ion and sediment fluxes from glacial catchments are controlled by runoff (e.g. see papers by some of the co-authors), and the authors acknowledge that the premise is not valid with respect to $\delta^{30}\text{Si}$ (lines 37-40). It is therefore rather awkward when the authors state that their study aim is to "challenge this assumption" (line 5).

As I see it, the most interesting aspect of the study is the $\delta^{30}\text{Si}$ measurements of Greenlandic meltwater. As noted by the authors, $\delta^{30}\text{Si}$ measurements of glacial meltwater are rare and the low $\delta^{30}\text{Si}$ values relative to non-glacial water are particularly intriguing. I would have preferred a thorough catchment-scale isotope study similar to the west Greenland isotope studies by Pogge von Strandmann and co-workers. The authors could have tracked down the light $\delta^{30}\text{Si}$ sources and provided new knowledge on subglacial secondary clay formation or subglacial soil erosion. As there is no new model development and the model results are based on numerous non-validated assumptions, a temporal and spatial extrapolation from the Greenland site could have been placed as the last sub-chapter in the Discussion chapter.

I found it difficult to follow the extrapolation from a few isotope samples collected at a glacier in Greenland to a global Si budget covering the last 21,000 years. As the cumulative uncertainty is huge and many assumptions are not validated, I end up having little trust in the model results. A critical assumption is that the Si concentrations and $\delta^{30}\text{Si}$ values from the Greenland glacier are representative for modern and palaeo-ice sheet runoff (lines 83-85 and 191-195). Since the entire study depends on this assumption, it must be tested quantitatively by analyzing meltwater samples from Antarctic, Greenlandic, Canadian, Icelandic and Norwegian glaciers. I acknowledge that it will be difficult to get samples from all relevant regions, but at least some samples from other glaciers should have been presented for validation. The lack of validating assumptions gives me the impression that the study is rushed.

I am also puzzled about the strategy behind the choice of study sites. Why examine a glacier in west Greenland and a marine sediment core from off southeast Iceland, when there are no obvious connections? The text does not provide a logical explanation. Why did the authors not analyze a sediment core from the Davis Strait, where ocean currents link the glacier runoff with the sediment core site? With respect to the Iceland core site, the local ocean current patterns should be explained to the reader. Would it not be more correct to use the $\delta^{30}\text{Si}$ values from Iceland (Georg et al., 2007; Opfergelt et al., 2013) for the comparison with the spicule $\delta^{30}\text{Si}$ values in the marine core from southeast Iceland? It is not clear to me whether the Si derives from the Gulf Stream or from a local Icelandic source. The consequences of changes in the ocean current pattern throughout the deglaciation and Holocene may have an impact on the source of Si to the sediment core site. This needs to be discussed.

Another thing that bothers me as a reader is that the findings are presented as novel, while the

true is that the findings just confirm previous findings. For example, it has been known for at least 10 years that the $\delta^{30}\text{Si}$ composition in glacial rivers is lower than non-glacial rivers (Georg et al., 2007, EPSL 261, 476-490), so why write that "our results show that these systems act as a significant source of isotopically light Si" (lines 255-256). Readers are left with the impression that this is the first study to show this, while this is not the case. There are too many "our" and "we" in the Discussion chapter. It will be more appropriate to use a neutral language and to focus on how the findings relate to previous findings.

Comments:

Title: The title does not reflect the focus on the North Atlantic. Delete "global" from the title to avoid overselling the manuscript. "Global" is not mentioned anywhere else in the text.

11: Typo in "uptake".

12: The study describes a period of deglaciation; not a period of expansion of ice sheets.

22-25: Is this only true, if Si is a limiting factor for diatom growth?

28-29: As fractionation due to secondary mineral formation is likely to dominate in a glacial catchment, it seems appropriate to elaborate on this process here.

37-38: This is a repetition of the premise of the study and again it is not supported by references.

60: "studies" -> more than one reference.

63 and 152: The study by Georg et al. (2007) also show that that glacial rivers have low $\delta^{30}\text{Si}$. It seems that the authors have overlooked this study from Iceland.

63-64: Why is it important that the glacial systems are large?

66: Delete "novel".

74: What is meant by "enhanced glacial activity"?

80: I assume that "summer" refers to June, July and August. In Figure 2A, it looks like the mean summer discharge is higher than c. 150 m³ s⁻¹, although most of the August record is missing.

80-81: It is unclear to me, what is meant by "and is an order of magnitude larger than others studied for Si to date". Are the authors referring to the glacier size or runoff? Obviously, Si concentrations has been quantified in glacial catchments that are larger in size and have higher annual runoff, so I assume that the authors mean $\delta^{30}\text{Si}$, not Si.

84: Why is glacier size important? The glacier covers 600 km² of a total GrIS area of 1710000 km². How can this glacier be representative when it comes to size? If it is close to the average catchment size for GrIS catchments, then add a reference to confirm this.

88-89: This extrapolation from Leverett Glacier to Greenland and the GrIS is not validated. The author should be specific when reporting data so that the data may not be misinterpreted or misused by other authors. Change "Greenland" to "Leverett Glacier" and change "GrIS" to "Leverett Glacier".

89: A consequence of using discharge-weighted mean concentrations is that all samples collected prior to day 170 have very limited impact on the reported mean concentrations. This basically

means that approximately half of all samples become irrelevant to the modelling and that the actual number of relevant samples become much lower. In the case of $\delta^{30}\text{Si}$ (ASi), $n = 6$ after day 170, which means that the statistical uncertainty is high.

89: What about Si concentrations and $\delta^{30}\text{Si}$ composition in August and September? I guess that they differ from the early melt-season because the subglacial drainage network is now well-developed. What effect does the lack of sampling in August and September have on the mean concentrations and mean $\delta^{30}\text{Si}$ composition? Is there a bias in the sampling here?

91: "similar" – these values do not appear to be similar to me. This statement must be supported by statistics.

92-93: "Meltwater outburst events ... in response to rapid supraglacial lake drainage ..." – The assumption that these four outbursts are caused by supraglacial lake drainage must be supported by evidence. Outbursts can be caused by other events such as intense melting on the ice sheet.

97: "collected the first measurements" – rephrase.

99: What is meant by "local bedrock"? Is it the bedrock at the sampling site or is it a mix of unconsolidated sediments, which are assumed to derive from the area beneath the glacier?

99-100: Insert the number of bedrock and bulk suspended sediment samples in the brackets after the values.

100: It looks like it is less than 0.2 in Figure 2D.

113: Comma before "respectively".

120: The flux should be in Mmol year^{-1} , not %, to make it consistent with the unit reported for the $\delta^{30}\text{Si}$, or am I wrong?

122: What is meant by "associated non-glacial riverine fluxes"?

124: I am very skeptical to extrapolate the time-scale to 21,000 years ago, as the oldest $\delta^{30}\text{Si}$ value in the Icelandic marine core is from c. 15,000 years ago (Figure 7C). This discrepancy is not clearly presented in the text, as it should, and the extrapolation is not validated by core correlation. Unless the authors come up with some solid arguments for the extrapolation, the model should only cover the period from 15,000 years ago to present.

131: It is only high-resolution for the period 10,000-15,000 years ago. The last 10,000 years are low-resolution ($n = 4$).

131: Why is it sufficient with just a single core off southeast Iceland to investigate changes in the global silicon cycle (as the title of the manuscript infers)?

136: The dating is likely to be very uncertain. The uncertainty in dating must be discussed here and horizontal error bars should be added to the data points in Figure 7C.

146: This is definitely something that would have been worthy to investigate before extrapolating the data from the catchment. What secondary siliceous minerals are present in the suspended sediment from the glacier?

169: Change to "interstadials and interglacials".

177-178: Delete "commonly referred to as outburst events".

183: Use the abbreviation GrIS.

189: Delete second comma.

191-195: These assumptions are critical and must be tested quantitatively.

197: What is the argument for applying Rayleigh-type kinetics instead of steady-state kinetics?

198-202: Here, I miss some text about other Si sources such as dust deposition, iceberg meltout and ocean current transport. What is known about the modern Si and $\delta^{30}\text{Si}$ of these sources and how have they changed during the deglaciation and Holocene?

200: Insert the values of the results in the text to make it easier for readers to compare them.

203: What is meant by "terrestrial weathering regime"?

203: How was the $\delta^{30}\text{Si}$ "weighted"? I assume that it should be "runoff-weighted". If so, mention how much of the variation can be explained by the change in runoff alone.

210: At least one reference is needed here.

222: Use the common term "carbon pump" instead of "silica pump".

234: Heinrich event 1 (H1) seems to have been ignored. Even if the authors assume that H1 has no influence on the Si sources in the marine core, the readers need to know about their arguments for neglecting H1. I suggest that the authors read Álvarez-Solas et al. (2011, *Clim. Past* 7, 1297-1306) and include H1 in their model.

243-246: Exactly. If the spicule $\delta^{30}\text{Si}$ coincides with collapse of the Icelandic palaeo-ice sheet, it will be more correct to use the Icelandic riverine $\delta^{30}\text{Si}$ values.

254-255: No, it has not been overlooked! Numerous papers have dealt with nutrient inputs from glacial rivers and icebergs to the marine environment.

274: Mention the uncertainty in the runoff measurements here.

291: Mention the number of samples here.

310-311: "dependent on the balance between ice sheet meltwater flux and riverine flux over the last 21,000 years" – This is another critical step in the modelling where several assumptions need to be addressed.

311: How about changes in the Si flux from iceberg meltout?

312: The Si flux has been quantified from several glaciers worldwide. How does the Si flux from Leverett Glacier compare to these other glaciers?

323: Use dust records from the Northern hemisphere instead of the Vostok dust record.

Fig. 2A. Change to "s" in the ordinate unit.

Fig. 3. The color code looks strange. Why is the area B orange when the rest of Greenland is dark green (sedimentary rocks)?

Fig. 7B. There should be a reference to these data.

SI page 2: The errors associated with discharge and suspended sediment measurements seem to be extremely low for a glacial catchment. See the work by Dirk van As to get an idea on how difficult it is to estimate discharge in large glacial rivers.

SI page 16: Update reference 16.

Reviewer #3 (Remarks to the Author):

Hawkings et al present some very interesting data regarding the possible role of ice sheets in the marine silicon cycle. They observe, for the first time, that the dissolved silicon in melt waters is extremely light such that the collapse of large paleo ice sheets may drive whole-ocean changes in the isotopic composition of dissolved Si in the sea towards lower values during the LGM. Lower $\delta^{30}\text{Si}$ values in diatoms from the LGM have been interpreted as reflecting changes in diatom productivity with the assumption that ocean $\delta^{30}\text{Si}(\text{OH})_4$ remains constant. If whole ocean $\delta^{30}\text{Si}(\text{OH})_4$ declined due to meltwater influence that interpretation needs to be re-evaluated.

The paper is well written and easy to follow. The data are sound. As Si isotope measurements are difficult I paid close attention to those measurements and conclude that they are of very high quality. I am a bit disappointed that only a single glacier was examined as this hinders generalization. While the runoff data are corroborated by the sponge spicule $\delta^{30}\text{Si}$ record in a nearby core, the authors admit that other smaller glaciers do not show this trend. So while the data are intriguing I am somewhat skeptical. However, if Nature views publishing new ideas as paramount, the authors present a transformational idea that merits publication and further study.

There are a few places in the paper where concepts are a bit muddled or in one case so oversimplified as to be of limited use. The calculations involved in the estimates of Si utilization during the last glacial maximum (lines 102-112) are based on equations that treat the entire ocean as one closed homogenous volume (supplemental information, page 5, equations 1-2). By using these equations it is assumed that the isotopic composition of the dissolved Si fueling diatom growth is that of freshwater (meltwater) input. This is a terrible assumption. In reality the Si supplied to the surface ocean arise from 100-300 m depth. The isotopic composition of this water is the real input term when evaluating productivity, i.e. utilization. I do agree that the isotopic composition of the 100-300m ocean water mixing to the surface would be modified by meltwater input, but the idea that this water has an isotopic signature of pure meltwater is wrong. Estimating the isotopic composition of these waters would take a model with adequate depth resolution.

The utility of a two- box model to estimate to evaluate impacts on the Si budget (line 114) is terribly oversimplified, but I appreciate that use of something more complex like box models used by Reynolds (2009) GBC are also fraught with assumptions when one is dealing with the past when water mass distributions may have been quite different. These are also easy models and the authors may want to try one to corroborate their model results.

A couple of minor points related to diatom biology and biogeochemistry:

Line 22: The idea that the input of Si from terrestrial weathering sustains diatoms assumes a certain timescale. Readers who work in the modern ocean would balk at the statement as written because diatom productivity would continue unabated for centuries or longer should all terrestrial Si input suddenly cease. Terrestrial inputs do counter losses through burial of opal in the sea so the statement is true on geological timescales related to the residence time of Si in the sea. I recommend being specific about the appropriate time frame.

Line 218. As written the reference to a 'silica pump' is such that one could interpret the sentence as meaning that the silica pump facilitates the export of organic material by diatoms. This is not true. The silica pump is a construct to explain how silicon can be preferentially removed from surface waters compared to nitrogen and carbon. A strong silica pump diminishes the importance of diatoms in the biological pump by essentially exporting empty frustules while C and N are recycled in surface waters. It is often invoked to explain the preferential drawdown of silicic acid relative to nitrate in HNLC regions.

Responses to reviewer on “The global silicon cycle impacted by past ice sheets”

Reviewer 1

We thank reviewer 1 for a thorough review of our paper and helpful suggestions for improving the manuscript. The main concern of the reviewer was our use of the Rayleigh distillation model of Opfergelt et al. (2013), and a two-box model, both of which we have revised in the new manuscript. The reviewer states that following these revisions the paper will make a major contribution to understanding nutrient cycling in the ocean. We have followed their advice, and substantially improved the modelling component, which we believe has led to a stronger paper.

1. “I cannot recommend the publication of this paper in its current form. This is mainly because I find the modelling tools that the authors have used to be inappropriate to the questions at hand, in one case fatally so...”

We have made substantial changes to the modelling component of the manuscript precisely as suggested by the reviewer. The Rayleigh model has been removed, and a 3-box model has replaced the previous 2-box model. See lines 116-183 and lines 261-341.

2. *“To a lesser degree, it is because I find one of the models insufficiently documented, and because I have questions regarding the interpretation of the marine sponge spicule $\delta^{30}\text{Si}$ record (although the authors may be able to easily answer these).”*

We have also revised and extended the documentation of the model and the model inputs in the Supplementary Information to make them clearer. The interpretation of the marine sponge spicule $\delta^{30}\text{Si}$ record is addressed below.

Major comments

1. *“Reconstruction of LGM/present Si utilisation*

The manuscript’s main claim that “diatom Si [uptake] during the LGM was likely similar to present day” (L10-11) is based on the model of Opfergelt et al. (2013), presented on L105-114 and in the Supplementary Information. Unfortunately, the authors of that study made some exceedingly poor assumptions in formulating their model, assumptions that the present authors have naturally taken over in their entirety. As a consequence, I cannot consider these results to be robust, or even meaningful, as I attempt to explain below.”

The section using the Rayleigh model of Opfergelt et al. (2013) has now been removed. These results are no longer presented in the manuscript. Thus we have followed the advice of the reviewer and made the necessary changes to address their concerns. The discussion and interpretation of palaeo data is now robust.

2. *“Modelling deglacial ocean change*

2a Choice of two-box model... Given this considerable effort, I am surprised that the authors have chosen to stick a two-box model of the ocean. I’m sure I don’t need to tell the authors that ever since 1984 (Knox and McElroy, 1984; Sarmiento and Toggweiler, 1984), it has been known that representation of glacial-interglacial ocean biogeochemistry requires consideration of the polar oceans, e.g. through use of a three-box model. The polar oceans are key in controlling the marine cycle of Si and its isotopes (Sarmiento et al., 2007; Brzezinski and Jones, 2015), and it is in the polar Southern Ocean that the clearest glacial-interglacial changes in marine diatom $\delta^{30}\text{Si}$ records are seen. Given the difference in surface Si concentrations between low- and high-latitudes in the modern ocean (and corresponding differences in the extent to which Si isotopes are fractionated), I would expect the deglacial isotopic response of a three-box model to be appreciably different to that of a two-box model. Whilst I appreciate the need for model simplicity over long integration times, models need to

be not only as simple as possible, but also as complex as necessary for the task at hand, and I believe a two-box model of the oceanic Si cycle does not fulfil this second requirement. And the additional computational cost of a three-box model would be easily bearable, in my opinion.”

The aim of the modelling component of this study was to provide a simple “thought experiment” and a means by which to test what would happen if we applied our Greenland data set to the glacial-interglacial transition. We have followed the reviewer’s recommendations and overhauled our modelling approach. We now present a three-box model.

3. *“A quantitative comparison between observational records and simulated polar ocean change, with special consideration of the timescale of this change, would be a real step forward in understanding the influence of external inputs on marine $\delta^{30}\text{Si}$ records. In contrast, the authors’ current approach limits them to showing that the modelled changes in input cause a change in mean-ocean $\delta^{30}\text{Si}$ of $\sim 0.25\text{‰}$ (ensemble mean) over the deglaciation. This is essentially the same result arrived at by Frings et al. (2016) using a similar two-box model, and thus not really novel.”*

Our new model results imply an ensemble model change of up to 0.18 ‰ (median of 0.13 ‰) in the Silicon isotopic composition of surface waters. Although this is not a significant departure from our previous estimate, we now provide a more robust approach. We argue that the novelty behind our results comes from the observed changes being driven purely by altering the Si supply to the oceans from riverine (glacial and non-glacial) inputs, with a focus on the much lighter Si delivery from glacial meltwaters.

4. *“2b. Model documentation: Another issue I have with the two-box model is that there is insufficient detail provided regarding how key behaviours are formulated; without these it remains entirely opaque to the reader. Given the variety of silica production behaviours modelled by De La Rocha and Bickle (2005), some of them rather ad hoc, it is not sufficient to simply refer to this paper for documentation.”*

We no longer use the two-box model of De La Rocha and Bickle. Details on the new three-box model are provided in lines 237-316 in the supplementary information.

5. *“It is vital for the reader to know what parameterisations were chosen for uptake and dissolution/burial, since these factors will be a key determinant of how the model’s whole-*

ocean Si inventory responds to changing inputs, as well as what might cause the higher DSi concentrations that the model simulates for the LGM ocean. Furthermore, the isotopic dynamics of the surface ocean are crucial in determining the ocean's isotopic response. This information should be supplied in the Supplementary Information.”

The three-box model is fully documented in the supplementary information. See lines 237-316.

6. “I have one main question regarding this sponge record, which shows remarkable variability. Could it possibly be that this variability is simply the result of the much higher resolution of this record relative to most sponge $\delta^{30}\text{Si}$ records, rather than actually reflecting greater variability at this site?... The Methods and SI provide no information on sampling, $\delta^{30}\text{Si}$ sub-sampling and sample preparation, so it is not possible to determine if, for instance, sample size is comparable between the high- and low-resolution records presented here.”

We argue that the $\delta^{30}\text{Si}$ signal is likely not being driven by the high resolution of this core. Spicule count data does not match the $\delta^{30}\text{Si}$ signal measured, thus the signal is not being driven by sample size. Core data from Hendry et al. (2016) and our core data indicates there is much less variability in the Holocene relative to the deglacial (see figure below). We argue that this difference indicates true variability during the deglacial compared to Holocene in our records. Our RAPiD 10-4P data is likely to be more variable than 15-4P because it's shallower, so more impacted by glacial meltwater.

7. “A second point that I think the authors should expand at least briefly on is the fact that their sediment core is from a water depth of ~1250m, whilst glacial runoff obviously

enters the ocean at its surface. Whilst the North Atlantic is a region of deep water formation today, my understanding is that northern ventilation was shallower during the first half of the deglaciation (and that meltwater pulses are associated with reduced ventilation). I think it would be helpful for the authors to comment at least briefly on the evidence for northern ventilation to the depths of their sediment core location, since this would significantly strengthen their argument.”

We agree with the reviewer that there was likely reduced deep water ventilation during the early part of the deglacial. However, this would likely have had minimal impact on shallow depths (<2 km). There was still localised polar influence from meltwaters, sea-ice formation and brine rejection that would have transferred a surface $\delta^{30}\text{Si}$ signal to depth via the overflow waters (Thornalley et al. 2011a, b). These overflow waters came from the Nordic Seas, which would have been heavily influenced by ice melt from northern hemisphere ice sheets. We have added some text to lines 359-364 to address this point.

8. *“Finally, comparing the Carolina Slope data shown in Fig. 7 with the data from this core published originally in Hendry et al. (2016), the data points appear to be shifted by a few kyrs. If the core’s age model has been revised, this should be mentioned somewhere; if not, it would appear that there is an error in Fig. 7.”*

We thank the reviewer for pointing this out. There was an error in Fig. 7 which has now been corrected.

Minor comments

1. *“Regarding the calculation of past Si utilisation, I would like to point out that the model the authors use has only ever been applied by Opfergelt et al. (2013), and not by De La Rocha et al. (1998) as the authors suggest on L110 and L303. Opfergelt et al. (2013) simply applied their model to data from De La Rocha et al. (1998), and arrived at different values for input to the global ocean based on two differing ocean Si budgets (see their Table 2). Note that De La Rocha et al. (1998), in order to produce a quantitative estimate of utilisation from their diatom $\delta^{30}\text{Si}$ data, assumed a source $\delta^{30}\text{Si}$ value of +1.6‰ (see their last paragraph) as an estimate of the isotopic composition of marine DSi being supplied to the euphotic zone from below.”*

We have now removed this component of the manuscript.

2. *“On L121, the authors state that the LGM-to-present change in the weighted $\delta^{30}\text{Si}$ of the input flux is +0.33‰, but the ensemble mean in Fig. 5c shows a change of around +0.48‰. Is there an error somewhere, or am I misunderstanding the authors?”*

The modelled component of the paper has been overhauled, including the input fluxes.

3. *“Fig. 6: from the figures it would appear that simulations are represented as anomalies from LGM conditions, and not from present-day conditions as stated in the caption (L643).”*

We thank the reviewer for pointing this out. The figure should read anomalies from LGM conditions. This has been corrected for the three-box model runs.

Reviewer 2

The reviewer raised some interesting points and important issues in their thorough review of the manuscript. We have addressed these in the revised manuscript and thank them for their useful comments. We believe some of these concerns arise from misunderstandings, which we have highlighted below and have clarified in the main text.

Major comments

1. *This study uses Si concentrations and Si isotope compositions from a glacier in west Greenland and a Si isotope record of sponge spicules in a marine sediment core off southern Iceland to model input Si flux and $\delta^{30}\text{Si}$ composition during the late Wisconsin and Holocene. The manuscript is interesting in the sense that it expands the previous work by De La Rocha, Treguer, Frings, Opfergelt, Georg and their co-workers, by focusing on the glacial Si component to the ocean Si budget. While this is a valid and praiseworthy study objective, I struggle to understand whether the findings of this study actually improve our current knowledge about the history of the global Si cycle significantly.*

Our study provides unique and significant findings that help frame the importance of glacial Si inputs during deglaciation events. We hope the responses to reviewer comments below and the changes to the manuscript help clarify the story, and demonstrate the importance of this study to a wider audience.

2. *The premise of the study is that “riverine silicon (Si) flux and isotope composition ($\delta^{30}\text{Si}$) are thought to be uniform in time and unaffected by the expansion and retreat of large ice sheets during glacial-interglacial cycles” (lines 2-4). I cannot recognize that this is the general understanding within the research community, and I am surprised that the authors believed so before this study was initiated in 2015*

We think there is a misunderstanding here. We were trying to say that previous authors thought there was no change in total riverine input on glacial-interglacial timescales. This is a matter of both composition and fluxes. Opfergelt et al. (2013) and Frings et al. (2016) started to address the composition question, but not a combination of composition and flux from rivers, while Georg et al. (2009) propose changes in the balance of groundwater vs riverine inputs. A changing flux and isotopic composition due to the expansion and retreat of ice sheets has not been addressed. We have clarified this in our revised manuscript – see lines 2-4.

3. *In fact, the premise contradicts the general understanding that ion and sediment fluxes from glacial catchments are controlled by runoff (e.g. see papers by some of the co-authors), and the authors acknowledge that the premise is not valid with respect to $\delta^{30}\text{Si}$ (lines 37-40). It is therefore rather awkward when the authors state that their study aim is to “challenge this assumption” (line 5).*

We believe this point arises from a misinterpretation of our manuscript. The papers in question address modern fluxes of nutrients from the Greenland Ice Sheet (e.g. Hawkings et al., 2015), and do not cover or address $\delta^{30}\text{Si}$ measurements at all, or mention changing fluxes between the LGM and Holocene. We do not understand the reviewer’s argument about the premise not being valid with respect to $\delta^{30}\text{Si}$. Lines 37-40 do not contradict this statement and no previous studies have addressed both the flux and $\delta^{30}\text{Si}$ composition change as a result of ice sheet expansion and deglaciation. The paper referenced here (Frings et al., 2016) is regularly cited throughout the manuscript and was the first study to hypothesise that changing riverine weathering regimes (but, importantly, not ice sheet fluxes and $\delta^{30}\text{Si}$ input) may drive some of the change. We expand on this by arguing that ice sheet inputs were likely more significant than previously thought, and provide an even lighter source of $\delta^{30}\text{Si}$ to the ocean (both in dissolved and amorphous form) than previously appreciated. For example, Opfergelt et al (2013) found a mean $\delta^{30}\text{Si}$ composition of $\sim 0.2\text{‰}$ from a few

spot measurements, whereas we find a mean composition $\sim 0.4\%$ lighter in ice sheet meltwaters.

4. *As I see it, the most interesting aspect of the study is the $\delta^{30}\text{Si}$ measurements of Greenlandic meltwater. As noted by the authors, $\delta^{30}\text{Si}$ measurements of glacial meltwater are rare and the low $\delta^{30}\text{Si}$ values relative to non-glacial water are particularly intriguing. I would have preferred a thorough catchment-scale isotope study similar to the west Greenland isotope studies by Pogge von Strandmann and co-workers. The authors could have tracked down the light $\delta^{30}\text{Si}$ sources and provided new knowledge on subglacial secondary clay formation or subglacial soil erosion. As there is no new model development and the model results are based on numerous non-validated assumptions, a temporal and spatial extrapolation from the Greenland site could have been placed as the last sub-chapter in the Discussion chapter.*

We believe this is exactly what we have done. This is the first ice sheet catchment study of $\delta^{30}\text{Si}$ in the literature. No other study to our knowledge has a $\delta^{30}\text{Si}$ time series this detailed or with such temporal coverage for such an environment. A large component of the manuscript deals with this issue. A more detailed multi-catchment weathering study warrants a separate manuscript and is currently in preparation.

5. *I found it difficult to follow the extrapolation from a few isotope samples collected at a glacier in Greenland to a global Si budget covering the last 21,000 years. As the cumulative uncertainty is huge and many assumptions are not validated, I end up having little trust in the model results.*

We acknowledge the uncertainties in this approach, although we maintain that our findings provide a first order and useful approximation of the changes that might be expected over glacial-interglacial studies. This is already documented in the manuscript text (e.g. lines 196-201), and provides the best estimate to date. We do our work to stimulate further debate and lead to more comprehensive estimates in the future. These uncertainties are also why we perform model sensitivity testing over a wide range of glacial input fluxes and $\delta^{30}\text{Si}$ compositions.

6. *A critical assumption is that the Si concentrations and $\delta^{30}\text{Si}$ values from the Greenland glacier are representative for modern and palaeo-ice sheet runoff (lines*

83-85 and 191-195). Since the entire study depends on this assumption, it must be tested quantitatively by analyzing meltwater samples from Antarctic, Greenlandic, Canadian, Icelandic and Norwegian glaciers. I acknowledge that it will be difficult to get samples from all relevant regions, but at least some samples from other glaciers should have been presented for validation. The lack of validating assumptions gives me the impression that the study is rushed.

The findings for Leverett Glacier are important because they are the first values reported for a large outlet glacier of an ice sheet system overlying representative shield bedrock, and the first time-series over a significant component of the summer ablation season (three months of data). These findings have important implications for our understanding of the Si cycle, and warrant publication in their own right. The composition of meltwaters from large ice sheet environments are very likely to be different from those draining much smaller valley glaciers (see Wadham et al., 2010, Hawkings et al., 2016 and Graly et al., 2014 for example). These data are extremely hard fought and by no means trivial to collect. It is not logistically easy to monitor a remote glacial catchment of this size for long periods. We agree with the reviewer that a global dataset would be useful, however, gathering samples from all these suggested environments is not a trivial (or economically feasible) task. With this paper we hope to further stimulate debate and research in this area, opening up the possibility of further studies.

7. *I am also puzzled about the strategy behind the choice of study sites. Why examine a glacier in west Greenland and a marine sediment core from off southeast Iceland, when there are no obvious connections? The text does not provide a logical explanation. Why did the authors not analyze a sediment core from the Davis Strait, where ocean currents link the glacier runoff with the sediment core site? With respect to the Iceland core site, the local ocean current patterns should be explained to the reader. Would it not be more correct to use the $\delta^{30}\text{Si}$ values from Iceland (Georg et al., 2007; Opfergelt et al., 2013) for the comparison with the spicule $\delta^{30}\text{Si}$ values in the marine core from southeast Iceland?*

We agree that, ideally, the glacial meltwater and core samples would be located closer together. Collection of core data is a major undertaking, and we are making use of previously collected materials, due to the numerous logistical and economical

challenges that would be involved with collecting a core in the Davis Strait (where to our knowledge no cores currently exist). The Iceland core provided an opportunity to retrieve a high-resolution archive of spicules over the appropriate time period, in order to test the hypothesis we present in the manuscript. The Icelandic meltwater data the reviewer mentions are from spot sampling events, and so do not represent a seasonally averaged value for interpretation – something we are able to fully investigate with our Greenland data. We would also like to reiterate a point made previously that the size of glacial catchment sampled also appears important in dictating geochemical composition of waters (Wadham et al., 2010), therefore sampling smaller glaciers in Iceland may also not be representative of the larger glacial catchments that would have existed with the Icelandic Ice Sheet.

8. *It is not clear to me whether the Si derives from the Gulf Stream or from a local Icelandic source. The consequences of changes in the ocean current pattern throughout the deglaciation and Holocene may have an impact on the source of Si to the sediment core site. This needs to be discussed.*

Previous interpretation of the core site suggests it is likely to reflect a Nordic Seas signal, so neither Iceland nor the Gulf Stream (Thornalley et al., 2011). This core site is much further north of the subpolar gyre and therefore does not reflect Si derived from the Gulf Stream.

9. *Another thing that bothers me as a reader is that the findings are presented as novel, while the true is that the findings just confirm previous findings. For example, it has been known for at least 10 years that the $\delta^{30}\text{Si}$ composition in glacial rivers is lower than non-glacial rivers (Georg et al., 2007, EPSL 261, 476-490), so why write that “our results show that these systems act as a significant source of isotopically light Si” (lines 255-256). Readers are left with the impression that this is the first study to show this, while this is not the case. There are too many “our” and “we” in the Discussion chapter. It will be more appropriate to use a neutral language and to focus on how the findings relate to previous findings.*

We thank the reviewer for pointing out the study by Georg et al. (2007) and have now included the reference in the manuscript. We have also removed a number of the

“our” and “we” in the Discussion chapter. The Opfergelt et al. (2013) study is cited widely in the manuscript (four times) and our values are compared to theirs. The phrase “our results show that these systems act as a significant source of isotopically light Si” is well justified, in that this is exactly what we show. Previous studies have not shown very light $\delta^{30}\text{Si}$ values associated with both high DSi and ASi. We agree that $\delta^{30}\text{Si}$ values from glacial meltwaters have previously been found to be lower, and this is acknowledged widely in the manuscript. However, we reiterate that our findings are novel. This is the first seasonally resolved glacial catchment dataset, the first dataset to show a $>1\%$ $\delta^{30}\text{Si}$ shift in dissolved Si in a single river, the first from a large ice sheet catchment in Greenland and the first to measure the $\delta^{30}\text{Si}$ of amorphous Si in suspended particulate matter. Furthermore, we show the lightest $\delta^{30}\text{Si}$ values found in running waters, $>0.5\%$ lighter than the lightest values reported by Opfergelt et al. (2013) and $>0.7\%$ lighter than the those reported by Georg et al. (2007).

Other comments

Title: The title does not reflect the focus on the North Atlantic. Delete “global” from the title to avoid overselling the manuscript. “Global” is not mentioned anywhere else in the text.

We provide evidence of global perturbations in our modelling study through ice sheet runoff from glacial to interglacial time periods. To avoid overselling these results we have removed “global” from the paper title.

11: Typo in “uptake”.

Corrected.

12: The study describes a period of deglaciation; not a period of expansion of ice sheets.

We describe a period from LGM to present day, focusing on the deglaciation. We have changed the wording to reflect the reviewer’s concerns.

22-25: Is this only true, if Si is a limiting factor for diatom growth?

We do not understand the reviewer’s point here. Rivers are the main source of Si to the oceans and diatoms have an absolute requirement for it. It does not matter if Si is limiting diatom growth or not for our statement to be true (in some oceanographic settings it is limiting).

28-29: As fractionation due to secondary mineral formation is likely to dominate in a glacial catchment, it seems appropriate to elaborate on this process here.

Secondary mineral formation is not likely to dominate in a glacial catchment. We talk largely about amorphous silica in the context of glacial sediments, which is likely to be a weathering product. We have added some text and an additional reference here.

37-38: This is a repetition of the premise of the study and again it is not supported by references.

It has been suggested previously that a) there have been changes in riverine input to the oceans on glacial-interglacial timescales and b) there have been changes in the balance of rivers vs. groundwaters (and so a change in the $\delta^{30}\text{Si}$ **of the total Si input** to the ocean), but no studies addressed a change in river $\delta^{30}\text{Si}$ composition until Frings et al. (2016) and Opfergelt et al. (2013), which are both cited here. We have added an additional reference here which hypothesised changes in $\delta^{30}\text{Si}$ during glacial-interglacial cycles could be driven by a change in weathering regime due to the presence of ice sheets (Georg et al., 2007), but this study made no quantitative estimates (this discussion makes up one paragraph in their manuscript). We maintain that no study has addressed a change in Si flux **and** $\delta^{30}\text{Si}$ composition driven by ice sheet runoff.

60: "studies" -> more than one reference.

We have added additional references here.

63 and 152: The study by Georg et al. (2007) also show that that glacial rivers have low $\delta^{30}\text{Si}$. It seems that the authors have overlooked this study from Iceland.

We thank the reviewer for pointing out this paper to us. We have now included this study and it is referenced multiple times in the manuscript.

63-64: Why is it important that the glacial systems are large?

Large catchments are more representative of glaciers discharging the majority of water from the Greenland Ice Sheet. Studies have shown that it is likely the geochemical composition of runoff is different between large ice sheet catchments and small valley glaciers (Wadham et al., 2010, Graly et al., 2014, Hawkings et al., 2016). We acknowledge that Leverett Glacier

(LG) covers a small proportion of the ice sheet. However, LG is an excellent representative study area for several reasons:

1. It is significantly greater (by almost two orders of magnitude) than any other glacial study area used thus far.
2. Its underlying geology and hydrology are thought to be representative of other large land terminating catchments around the Greenland Ice Sheet, as documented in other studies (Bartholomew et al., 2011; Hawkings et al., 2015)
 - a. Discharge is proportional to total modelled ice sheet runoff for the years studied (Hawkings et al., 2015).
 - b. The underlying debris and morphology of the catchment are similar to >75% of the West Greenland ice margin (Knight et al., 2002)
 - c. Bedrock geology is predominantly Neoproterozoic gneiss/granite, which is typical of large areas of the crystalline rocks that dominate the Precambrian Shield on which Greenland lies (Henriksen et al., 2009; Kalsbeek, 1982).
 - d. Upscaling suspended sediment loads give a value within the range quoted by a recent Greenland wide satellite study (Overeem et al., 2017)

Catchment hydrology is well documented and, although comparative datasets are thus far lacking, is believed typical of the large Greenland outlet glaciers that dominate discharge of meltwaters from the Greenland Ice Sheet (Chandler et al., 2013; Chu, 2014; Cowton et al., 2013; Tedstone et al., 2013).

We have now expanded our explanation in discussion section of why Leverett Glacier is an excellent study area for this kind of work.

66: Delete “novel”.

Done.

74: What is meant by “enhanced glacial activity”?

We hoped it would be read as representing periods of greater glacial coverage – e.g. ice ages.

We have changed the phrasing to reflect the reviewer’s concerns.

80: I assume that “summer” refers to June, July and August. In Figure 2A, it looks like the mean summer discharge is higher than c. 150 m³ s⁻¹, although most of the August record is missing.

Summer refers to May-September. We have changed the terminology to make this clearer. Our discharge record lasts until the middle of September, however, this is outside our geochemical monitoring period, so we have not recorded it in Figure 2A. We have now included the full discharge record in the supplementary information.

80-81: It is unclear to me, what is meant by “and is an order of magnitude larger than others studied for Si to date”. Are the authors referring to the glacier size or runoff? Obviously, Si concentrations has been quantified in glacial catchments that are larger in size and have higher annual runoff, so I assume that the authors mean $\delta^{30}\text{Si}$, not Si.

We respectfully disagree with the reviewer’s comments here. Leverett Glacier is at least an order of magnitude larger in size than other glaciers studied in this context to date. If the reviewer knows of another study where they have monitored a larger ice sheet catchment ($>600 \text{ km}^2$ with mean summer discharge of $\sim 150 \text{ m}^3 \text{ sec}^{-1}$) for Si concentrations and $\delta^{30}\text{Si}$ composition and would direct us to it, then we would be happy to incorporate the findings into our study design and data interpretation (this is by no means obvious). A study (Yde et al., 2014) has measured DSi concentrations in the Watson River (to which Leverett Glacier feeds). However, this river has a combined discharge of three glacial systems and only DSi concentrations were measured (not ASi measurements or $\delta^{30}\text{Si}$ composition). In this study four DSi values are presented in Figure 1. Thus, the study is not comparable.

84: Why is glacier size important? The glacier covers 600 km² of a total GrIS area of 1710000 km². How can this glacier be representative when it comes to size? If it is close to the average catchment size for GrIS catchments, then add a reference to confirm this.

Please see the response and references we posted to the reviewer’s comment above.

88-89: This extrapolation from Leverett Glacier to Greenland and the GrIS is not validated. The author should be specific when reporting data so that the data may not be misinterpreted or misused by other authors. Change “Greenland” to “Leverett Glacier” and change “GrIS” to “Leverett Glacier”.

This is not an extrapolated value. The value presented is the flux from Leverett Glacier in 2015. We have made some edits here to make this clearer.

89: A consequence of using discharge-weighted mean concentrations is that all samples collected prior to day 170 have very limited impact on the reported mean concentrations. This basically means that approximately half of all samples become irrelevant to the modelling and that the actual number of relevant samples become much lower. In the case of $\delta^{30}\text{Si}$ (ASi), $n = 6$ after day 170, which means that the statistical uncertainty is high.

We infer from this comment that the reviewer agrees with our approach to apply a discharge weighted mean to our data. The point of using a discharge weighted mean is to place less importance on values where discharge is low. We want to obtain a mean value most representative of an annual “average” flux. This does not detract from the fact that early season/low discharge data are still interesting, because they allow us to track the annual evolution of subglacial drainage pathways and geochemical conditions at the glacier bed. Statistical uncertainty is high, which is why we incorporate sensitivity testing in our modelling using a variety of potential conditions. We wish to emphasise that these are the first $\delta^{30}\text{Si}$ measurements of ASi, and that the values we report (over the entire period) show very little deviation from one another, giving us confidence that the composition does not likely change significantly over the time period.

89: What about Si concentrations and $\delta^{30}\text{Si}$ composition in August and September? I guess that they differ from the early melt-season because the subglacial drainage network is now well-developed. What effect does the lack of sampling in August and September have on the mean concentrations and mean $\delta^{30}\text{Si}$ composition? Is there a bias in the sampling here?

The reviewer points out the difficulty in measuring such rivers. Three months of sampling in such remote environments is time consuming, logistically challenging and expensive both economically and in terms of the personnel needed to run such camps. We did our best to cover a significant proportion of the melt season but acknowledge that we lack August and September values. We would like to emphasise that collection of such datasets is not trivial and this presents a significant improvement on data previously published from glacial catchments.

91: “similar” – these values do not appear to be similar to me. This statement must be supported by statistics.

We have rephrased this sentence to reflect the reviewer’s concerns.

92-93: *“Meltwater outburst events ... in response to rapid supraglacial lake drainage ...” – The assumption that these four outbursts are caused by supraglacial lake drainage must be supported by evidence. Outbursts can be caused by other events such as intense melting on the ice sheet.*

Previous work has highlighted the importance of supraglacial lake drainage events in driving outburst events at Leverett Glacier (e.g. Bartholomew et al. 2011), but we acknowledge this phrase might appear misleading. We have therefore removed the reference to supraglacial lake drainage here and added a reference.

97: *“collected the first measurements” – rephrase.*

These are the first measurements of the $\delta^{30}\text{Si}$ of ASi so we believe this phrase is justified.

99: *What is meant by “local bedrock”? Is it the bedrock at the sampling site or is it a mix of unconsolidated sediments, which are assumed to derive from the area beneath the glacier?*

Local bedrock is unconsolidated glacial till previously overridden by the glacier, collected near the sampling site and assumed to represent the mixture of bedrock. This is already detailed in the supplementary materials and methods.

99-100: *Insert the number of bedrock and bulk suspended sediment samples in the brackets after the values.*

Done.

100: *It looks like it is less than 0.2 in Figure 2D.*

It is approximately 0.2, thus we include “~” before the value. We have changed this to ~0.1-0.2 ‰ to reflect the reviewers concerns.

113: *Comma before “respectively”.*

This section has now been deleted to reflect the concerns of reviewer 1.

120: *The flux should be in Mmol year⁻¹, not %, to make it consistent with the unit reported for the $\delta^{30}\text{Si}$, or am I wrong?*

The reviewer has misunderstood. The change is a relative change of - 11 ‰. $\delta^{30}\text{Si}$ is expressed as per mil (‰) not as %. This section has been significantly changed to reflect the concerns of Reviewer 1 and the Editor.

122: What is meant by “associated non-glacial riverine fluxes”?

Riverine fluxes are not glacial in origin – i.e. rivers not fed predominantly by glaciers. This section has been heavily edited to reflect the change to the new three-box model.

124: I am very skeptical to extrapolate the time-scale to 21,000 years ago, as the oldest $\delta^{30}\text{Si}$ value in the Icelandic marine core is from c. 15,000 years ago (Figure 7C). This discrepancy is not clearly presented in the text, as it should, and the extrapolation is not validated by core correlation. Unless the authors come up with some solid arguments for the extrapolation, the model should only cover the period from 15,000 years ago to present.

The reviewer has not fully appreciated the use of the sediment core data in this study. The model is not included in the manuscript to test the marine sediment core. The core data merely provides some constraints on the deglacial portion of the model, while providing additional evidence on which to test our hypothesis. The model is a thought experiment to help conceptualise the impact on glacial-interglacial timescales that glacially sourced light dissolved and particulate silica might have.

131: It is only high-resolution for the period 10,000-15,000 years ago. The last 10,000 years are low-resolution ($n = 4$).

Please see our comment above. We would like to point the reviewer toward previous core records for spicule and diatom $\delta^{30}\text{Si}$ composition (see Hendry et al., 2016 for example), and the challenges of generating such records. Further, the high-resolution record captures the main portion of the deglaciation when rapid sea level changes occurred, which is the period we are most interested in.

131: Why is it sufficient with just a single core off southeast Iceland to investigate changes in the global silicon cycle (as the title of the manuscript infers)?

We also present previously published Carolina Slope core data (see Figure 7) and discuss $\delta^{30}\text{Si}$ results from other cores in ocean basins around the world in the manuscript. We reiterate that such core records are rare and difficult to obtain.

136: The dating is likely to be very uncertain. The uncertain in dating must be discussed here and horizontal error bars should be added to the data points in Figure 7C.

Quantifying the errors in an age model is not valid as it is based on a mixture of timepoints (radiocarbon, formaminiferal abundance, d18O...), and the age model has been previously published (Thornalley et al., 2011). A robust error analysis would involve Monte Carlo simulations of uncertainties, which is out of the scope of the requirements for this paper.

146: This is definitely something that would have been worthy to investigate before extrapolating the data from the catchment. What secondary siliceous minerals are present in the suspended sediment from the glacier?

We refer the reviewer to our previous work on this – Hawkings et al. (2017). There is still much debate as to the formation of inorganic ASi, and we present the most likely hypothesis here, based on published data.

169: Change to “interstadials and interglacials”.

Done

177-178: Delete “commonly referred to as outburst events”.

Done

183: Use the abbreviation GrIS.

Done

189: Delete second comma.

Done

191-195: These assumptions are critical and must be tested quantitatively.

We refer the reviewer to our comments on the representativeness of Leverett catchment above. We agree that this could be tested further (hence we use the term “crude analogue”), but our argument is that it’s a reasonable assumption that LG is representative of large scale ice sheet systems (as we have previously detailed).

197: What is the argument for applying Rayleigh-type kinetics instead of steady-state kinetics?

This section has been removed in the revised manuscript.

198-202: Here, I miss some text about other Si sources such as dust deposition, iceberg meltout and ocean current transport. What is known about the modern Si and $\delta^{30}\text{Si}$ of these sources and how have they changed during the deglaciation and Holocene?

This discussion has been largely covered by Frings et al. (2016), which we reference in the text. The variables chosen are fully described in an expanded supplementary information. There are many uncertainties in the dataset, and the subject is relatively young, as Frings et al. point out. A full discussion of these values would repeat the discussion in Frings et al. (2016). This manuscript focuses on the glacial riverine inputs and the relative changes in non-glacial riverine inputs.

200: Insert the values of the results in the text to make it easier for readers to compare them

We have removed this section from the manuscript as suggested by one of the other reviewers.

203: What is meant by “terrestrial weathering regime”?

Removed from the manuscript as above.

203: How was the $\delta^{30}\text{Si}$ “weighted”? I assume that it should be “runoff-weighted”. If so, mention how much of the variation can be explained by the change in runoff alone.

This should have been noted as runoff-weighted. This section has been removed from the manuscript as above.

210: At least one reference is needed here.

Done.

222: Use the common term “carbon pump” instead of “silica pump”.

The term “silica pump” is distinct from the carbon pump, and we cite papers that originally describe this concept (e.g. Dugdale et al., 1995). To address the concerns of the reviewer we have revised and expanded this sentence – see lines 323-326.

234: Heinrich event 1 (H1) seems to have been ignored. Even if the authors assume that H1 has no influence on the Si sources in the marine core, the readers need to know about their arguments for neglecting H1. I suggest that the authors read Álvarez-Solas et al. (2011, Clim. Past 7, 1297-1306) and include H1 in their model.

We focus on meltwater pulses, as these are the timings of significant sea-level rise during the deglaciation (and so total input of freshwater to the oceans). Heinrich Stadial 1 and the Younger Dryas did not experience discernible sea-level rises despite being periods of deepwater formation “slowdowns/shutdowns”. Meltwater location, rather than total input, appears to have been important for HS1 (and YD), which is of course important, but not key to the main hypothesis here and is beyond the scope of the manuscript. See Stanford et al. (2006) for further discussion. We would like to emphasise that the modelling component of the manuscript is a thought experiment, to test the possible impact of glacial runoff on the marine Si cycle, rather than the focus of the manuscript.

243-246: Exactly. If the spicule $\delta^{30}\text{Si}$ coincides with collapse of the Icelandic palaeo-ice sheet, it will be more correct to use the Icelandic riverine $\delta^{30}\text{Si}$ values.

We would like to refer the reviewer to our previous points. There are several good reasons why we do not do this. These are point measurements (i.e. are not temporally resolved despite the large changes that occur seasonally in glacial systems), contain no information on ASi concentrations or isotopic composition, and are from smaller valley glaciers – not from a large ice sheet system, which is likely to behave differently (Wadhwa et al., 2010). We force the model with 50 scenarios picked using a Latin Hypercube sampling matrix. The model is not being used to validate core data or vice-versa. The core data is a discrete part of the manuscript and provides evidence of isotopic variation during periods of high meltwater input into the northern oceans.

254-255: No, it has not been overlooked! Numerous papers have dealt with nutrient inputs from glacial rivers and icebergs to the marine environment.

We have changed this sentence and added a reference to reflect the reviewer’s concerns.

274: Mention the uncertainty in the runoff measurements here.

Done.

291: *Mention the number of samples here.*

Done.

310-311: *“dependent on the balance between ice sheet meltwater flux and riverine flux over the last 21,000 years” – This is another critical step in the modelling where several assumptions need to be addressed.*

This is fully addressed in an expanded supplementary information.

311: *How about changes in the Si flux from iceberg meltout?*

The reviewer has highlighted an interesting point here. Iceberg contributions during the deglaciation are notoriously tricky to quantify (both freshwater and sediment flux), require a fully resolved ice sheet model, and, as such, values are hugely uncertain. If we added this component to our estimates it would purely be a guess at present. Quantifying iceberg fluxes, particularly of ASi, is a natural extension for future studies.

312: *The Si flux has been quantified from several glaciers worldwide. How does the Si flux from Leverett Glacier compare to these other glaciers?*

Meire et al. (2016) present a table of Si fluxes from other glacial catchments in Greenland, however these do not include ASi estimates and are solely DSi fluxes. There is one catchment (Saqqap Sermersua) which is notable for its very high flux compared to Leverett Glacier (LG 2015 flux = $331 \times 10^6 \text{ mol yr}^{-1}$, Saqqap Sermersua = $660 \times 10^6 \text{ mol yr}^{-1}$), but the estimate is based on a single sample, and modelled meltwater runoff values (i.e. not measured, which is problematic). Their single spot measurement value for DSi ($33 \mu\text{M}$) is higher than our mean concentration ($20.8 \mu\text{M}$) and within our measured range ($9.2 - 56.9 \mu\text{M}$). One glacier Kuannersuit Kuussuat has anomalously high DSi (mean = $280 \mu\text{M}$), but the river is based on basaltic bedrock (Disko Island), drains a surging glacier, and appears an outlier. We have previously measured an assortment of other runoff samples from this location (5 glaciers – manuscript in review) and all had DSi concentrations $<35 \mu\text{M}$.

323: *Use dust records from the Northern hemisphere instead of the Vostok dust record.*

We have changed our modelling approach in the new manuscript. The aeolian dust input has been kept constant in the new model, so that solely changes in the riverine input are assessed using the model.

Fig. 2A. Change to “s” in the ordinate unit

Done

Fig. 3. The color code looks strange. Why is the area B orange when the rest of Greenland is dark green (sedimentary rocks)?

The rest of Greenland is not dark green and the effect is probably due to the colour used to mark the land/ocean boarder. Some parts of northern Greenland have sedimentary rocks but the majority is shield bedrock. We have updated the map to make this clearer.

Fig. 7B. There should be a reference to these data.

Reference added.

SI page 2: The errors associated with discharge and suspended sediment measurements seem to be extremely low for a glacial catchment. See the work by Dirk van As to get an idea on how difficult it is to estimate discharge in large glacial rivers.

We appreciate these concerns in principle, but reject the suggestion that we do not have ‘an idea on how difficult it is to estimate discharge in large glacial rivers’. Monitoring of the Leverett Glacier catchment commenced in 2009 and has been led by glacio-hydrological experts with substantial field experience in the European Alps and Canadian High Arctic on decadal timescales. There is now a wide range of literature which discusses the hydrology of the Leverett Glacier catchment in detail, see this link for a synopsis and references:

<http://data.bas.ac.uk/metadata.php?id=GB/NERC/BAS/PDC/00841>

As we note in the SI, derivation of discharge errors is explained at length in Bartholomew et al. and Cowton et al. (2011; 2012), and we follow their approach here to calculate 2015-specific values of RMSD: $52 \text{ m}^3 \text{ s}^{-1}$, and normalised RMSD: 12.1%. These values are broadly equivalent to the errors calculated by van As et al. (2017), which are on the order of 20% uncertainty for the bulk melt season runoff (see their Table 1).

Regarding calculation of suspended sediment measurement errors, we note that this is not a feature of van As et al’s work. Our error estimates here were calculated following Cowton et al. (2012) which contains a fuller critique of the suspended sediment measurement setup at Leverett Glacier.

In summary, we are confident of the integrity of these datasets.

SI page 16: Update reference 16.

Done.

Reviewer 3

Although the third reviewer was generally positive about the manuscript, and highlighted the quality and importance of the data, some concerns were raised, such as the interpretations of silica utilisation using the model presented by Opfergelt et al. (2013). We have addressed these points below and in the revised manuscript.

Hawkings et al present some very interesting data regarding the possible role of ice sheets in the marine silicon cycle. They observe, for the first time, that the dissolved silicon in melt waters is extremely light such that the collapse of large paleo ice sheets may drive whole-ocean changes in the isotopic composition of dissolved Si in the sea towards lower values during the LGM. Lower $\delta^{30}\text{Si}$ values in diatoms from the LGM have been interpreted as reflecting changes in diatom productivity with the assumption that ocean $\delta^{30}\text{Si}(\text{OH})_4$ remains constant. If whole ocean $\delta^{30}\text{Si}(\text{OH})_4$ declined due to meltwater influence that interpretation needs to be re-evaluated.

We thank the reviewer for these comments and are glad they see the benefit of this study to the scientific community.

The paper is well written and easy to follow. The data are sound. As Si isotope measurements are difficult I paid close attention to those measurements and conclude that they are of very high quality. I am a bit disappointed that only a single glacier was examined as this hinders generalization.

Large catchments are more representative of glaciers discharging most of the water from the Greenland Ice Sheet. Studies have shown that it is likely the geochemical composition of runoff is different between large ice sheet catchments and small valley glaciers (Graly et al., 2014; Hawkings et al., 2016; Wadham et al., 2010). We acknowledge that Leverett Glacier (LG) covers a small proportion of the ice sheet. However, LG is an excellent representative study site at which to conduct such a study. We think the data presented here justifies publication in its own right, and believe that it will stimulate further study and debate.

While the runoff data are corroborated by the sponge spicule $\delta^{30}\text{Si}$ record in a nearby core, the authors admit that other smaller glaciers do not show this trend.

Smaller glaciers still show a trend toward lighter $\delta^{30}\text{Si}$ than average non-glacial riverine values, but not to the same extent – for example Opfergelt et al. (2013) find a mean $\delta^{30}\text{DSi}$ of $+0.17 \pm 0.18$ ‰ and Georg et al. (2007) $+0.63 \pm 0.38$ ‰. Neither study have measurements of ASi (concentration or isotopic composition). We have made this clearer in the revised manuscript.

So while the data are intriguing I am somewhat skeptical. However, if Nature views publishing new ideas as paramount, the authors present a transformational idea that merits publication and further study.

Please see point above. We also find distinctive light $\delta^{30}\text{Si}$ in other smaller glacial systems from around the world, the results of which will constitute a future publication.

There are a few places in the paper where concepts are a bit muddled or in one case so oversimplified as to be of limited use. The calculations involved in the estimates of Si utilization during the last glacial maximum (lines 102-112) are based on equations that treat the entire ocean as one closed homogenous volume (supplemental information, page 5, equations 1-2). By using these equations it is assumed that the isotopic composition of the dissolved Si fueling diatom growth is that of freshwater (meltwater) input. This is a terrible assumption. In reality the Si supplied to the surface ocean arise from 100-300 m depth. The isotopic composition of this water is the real input term when evaluating productivity, i.e. utilization. I do agree that the isotopic composition of the 100-300m ocean water mixing to the surface would be modified by meltwater input, but the idea that this water has an isotopic signature of pure meltwater is wrong. Estimating the isotopic composition of these waters would take a model with adequate depth resolution.

We have now removed this component of the manuscript.

The utility of a two- box model to estimate to evaluate impacts on the Si budget (line 114) is terribly oversimplified, but I appreciate that use of something more complex like box models used by Reynolds (2009) GBC are also fraught with assumptions when one is dealing with the past when water mass distributions may have been quite different. These are also easy models and the authors may want to try one to corroborate their model results.

We have overhauled the modelling component of our manuscript. We now use a slightly more complex 3-box model. Please see lines 116-185 and 274-331.

A couple of minor points related to diatom biology and biogeochemistry:

Line 22: The idea that the input of Si from terrestrial weathering sustains diatoms assumes a certain timescale. Readers who work in the modern ocean would balk at the statement as written because diatom productivity would continue unabated for centuries or longer should all terrestrial Si input suddenly cease. Terrestrial inputs do counter losses through burial of opal in the sea so the statement is true on geological timescales related to the residence time of Si in the sea. I recommend being specific about the appropriate time frame.

We agree with the reviewer and have changed this sentence to reflect this concern – added “over oceanic residence times”.

Line 218. As written the reference to a ‘silica pump’ is such that one could interpret the sentence as meaning that the silica pump facilitates the export of organic material by diatoms. This is not true. The silica pump is a construct to explain how silicon can be preferentially removed from surface waters compared to nitrogen and carbon. A strong silica pump diminishes the importance of diatoms in the biological pump by essentially exporting empty frustules while C and N are recycled in surface waters. It is often invoked to explain the preferential drawdown of silicic acid relative to nitrate in HNLC regions.

We have changed this section to address the reviewer’s concerns. Please see lines 319-322.

References

Bartholomew, I., Nienow, P., Sole, A., Mair, D., Cowton, T., Palmer, S., Wadham, J. (2011) Supraglacial forcing of subglacial drainage in the ablation zone of the Greenland ice sheet. *Geophysical Research Letters* 38, L08502.

Chandler, D.M., Wadham, J.L., Lis, G.P., Cowton, T., Sole, A., Bartholomew, I., Telling, J., Nienow, P., Bagshaw, E.A., Mair, D., Vinen, S., Hubbard, A. (2013) Evolution of the subglacial drainage system beneath the Greenland Ice Sheet revealed by tracers. *Nature Geoscience* 6, 195-198.

Chu, V.W. (2014) Greenland ice sheet hydrology: A review. *Progress in Physical Geography* 38, 19-54.

Cowton, T., Nienow, P., Bartholomew, I., Sole, A., Mair, D. (2012) Rapid erosion beneath the Greenland ice sheet. *Geology* 40, 343-346.

Cowton, T., Nienow, P., Sole, A., Wadham, J., Lis, G., Bartholomew, I., Mair, D., Chandler, D. (2013) Evolution of drainage system morphology at a land-terminating Greenlandic outlet glacier. *Journal of Geophysical Research-Earth Surface* 118, 29-41.

Graly, J.A., Humphrey, N.F., Landowski, C.M., Harper, J.T. (2014) Chemical weathering under the Greenland Ice Sheet. *Geology*, G35370. 35371.

Hawkings, J., Wadham, J., Tranter, M., Telling, J., Bagshaw, E., Beaton, A., Simmons, S.-L., Chandler, D., Tedstone, A., Nienow, P. (2016) The Greenland Ice Sheet as a hotspot of phosphorus weathering and export in the Arctic. *Global Biogeochemical Cycles* 30, 191-210.

Hawkings, J.R., Wadham, J.L., Tranter, M., Lawson, E., Sole, A., Cowton, T., Tedstone, A.J., Bartholomew, I., Nienow, P., Chandler, D., Telling, J. (2015) The effect of warming climate on nutrient and solute export from the Greenland Ice Sheet. *Geochemical Perspectives Letters* 1, 94-104.

Henriksen, N., Higgins, A.K., Kalsbeek, F., Pulvertaft, T.C.R. (2009) Greenland from Archaean to Quaternary Descriptive text to the 1995 Geological map of Greenland, 1:2 500 000. 2nd edition. *Geological Survey of Denmark and Greenland Bulletin*, 9-116.

Kalsbeek, F. (1982) The evolution of the Precambrian Shield of Greenland. *Geologische Rundschau* 71, 38-60.

Knight, P.G., Waller, R.I., Patterson, C.J., Jones, A.P., Robinson, Z.P. (2002) Discharge of debris from ice at the margin of the Greenland ice sheet. *Journal of Glaciology* 48, 192-198.

Overeem, I., Hudson, B.D., Syvitski, J.P.M., Mikkelsen, A.B., Hasholt, B., van den Broeke, M.R., Noël, B.P.Y., Morlighem, M. (2017) Substantial export of suspended sediment to the global oceans from glacial erosion in Greenland. *Nature Geoscience* 10, 859.

Tedstone, A.J., Nienow, P.W., Sole, A.J., Mair, D.W.F., Cowton, T.R., Bartholomew, I.D., King, M.A. (2013) Greenland ice sheet motion insensitive to exceptional meltwater

forcing. *Proceedings of the National Academy of Sciences of the United States of America* 110, 19719-19724.

Thornalley, D.J.R., Elderfield, H., McCave, I.N. (2011) Reconstructing North Atlantic deglacial surface hydrography and its link to the Atlantic overturning circulation. *Global and Planetary Change* 79, 163-175.

van As, D., Bech Mikkelsen, A., Holtegaard Nielsen, M., Box, J.E., Claesson Liljedahl, L., Lindbäck, K., Pitcher, L., Hasholt, B. (2017) Hypsometric amplification and routing moderation of Greenland ice sheet meltwater release. *The Cryosphere* 11, 1371-1386.

Wadham, J.L., Tranter, M., Skidmore, M., Hodson, A.J., Prisco, J., Lyons, W.B., Sharp, M., Wynn, P., Jackson, M. (2010) Biogeochemical weathering under ice: Size matters. *Global Biogeochemical Cycles* 24, GB3025.

Reviewers' comments:

Reviewer #2 (Remarks to the Author):

I was reviewer #2 in the last round of reviewing. In my opinion, the authors have done a good job in addressing my comments and improving the manuscript. In particular, I prefer the new modeling exercise. It is still my opinion that the riverine $\delta^{30}\text{Si}$ data is interesting and worth publishing, but I have to admit that I am still skeptical to the study design.

I was critical to the premise of the study that "until recently, it was assumed that riverine $\delta^{30}\text{Si}$ input was uniform over glacial-interglacial timescales" (38-39). The authors removed this postulate from the Abstract but chose to keep it in the main text (38-39). In their response, the authors highlight studies that examine $\delta^{30}\text{Si}$ input, but they still fail to show who "assumed that $\delta^{30}\text{Si}$ input was uniform" and the statement remains unsupported by literature references. Just because trends in a flux or a composition have not been investigated, it cannot be concluded that the understanding in the research community is or was that fluxes and compositions have not varied over time. In fact, I will claim that the general understanding is that riverine ion fluxes and isotope compositions (e.g., $\delta^7\text{Li}$, $\delta^{26}\text{Mg}$, $\delta^{44}\text{Ca}$, $\delta^{56}\text{Fe}$) varied from LGM to today because observations from natural low-temperature environments today show that these fluxes and compositions vary (of course I cannot support my claim with literature references). I hope that the authors now better understand my argumentation and my recommendation to rephrase this sentence.

I still find the study design strange, and the authors have not revised the manuscript to clarify their thoughts behind selecting a field site in west Greenland and a marine core off southern Iceland to the readers. For a study published in a Nature journal, I would expect researchers to optimize the study design to provide the best results possible, although I of course respect the logistical and economical costs that can be financed by a research proposal. In its current form, the study seems opportunistic.

The authors have chosen to keep the statement that "Leverett Glacier ... is an order of magnitude larger than other glaciers studied for Si to data", and in the rebuttal letter the authors restate that "Leverett Glacier is at least an order of magnitude larger in size than other glaciers studied in this context to date". First, the sentence refers to "glaciers studied for Si", not glaciers studied for both "Si concentrations and $\delta^{30}\text{Si}$ composition", as the authors seem to indicate in their response to my comment. Secondly, I was actually thinking of Skeidararjökull, which would be the obvious choice for a field site, as this outlet glacier is located close to the offshore core site and drains towards the south. The catchment area of Skeidararjökull is generally estimated to approximately 1380 km², or more than twice the area of the glacier used in this study. In addition to published spot sampling, Gíslason et al. (Chem. Geol. 190, 2002) presented a time-series of Si concentrations and Opfegelt et al. (2013) presented a $\delta^{30}\text{Si}$ value. Thirdly, it is not obvious for readers to know when the authors think that a study qualifies as a study "studied for Si" if a presentation of Si concentrations is not enough. How should readers know that a study that "has a combined discharge of three glacial systems" (rebuttal letter) or studies from the Himalayas (e.g., Batura or Gangotri glaciers) or studies from Antarctica are disqualified? The sentence does not mention anything about that the studies must be comparable to Leverett Glacier in all aspects.

As Skeidararjökull is significantly larger than the Greenlandic field site, I am confused by some of the authors' responses such as: "We would also like to reiterate a point made previously that the size of glacial catchment sampled also appears important in dictating geochemical composition of waters (Wadham et al., 2010), therefore sampling smaller glaciers in Iceland may also not be representative of the larger glacial catchments that would have existed with the Icelandic Ice Sheet". I would have liked to see Si and $\delta^{30}\text{Si}$ time-series from Skeidararjökull included in the study in combination with time-series from a granitic site such as the one from Greenland. After all, this is submission to a Nature journal.

Other comments:

42, 46: Palaeo- is a prefix. Insert hyphens.

91: Thanks for pointing my attention to Hawkings et al. (2017). However, the authors oversell the representativeness of Si fluxes from Leverett Glacier in 2015 here. As I understand Hawkings et al. (2017), "previously reported for the GrIS" actually refers to "previously reported for Leverett Glacier in 2012", as the comparison is only between fluxes in runoff.

91-92: The readers should know more about the interannual variations at Leverett Glacier. Annual differences in DSi concentrations of 9.6 and 20.8 μM seem large and need to be presented for the reader and discussed in more detail in relation to the model assumptions.

100, 104, elsewhere: The notations $\delta^{30}\text{ASi}$ and $\delta^{30}\text{DSi}$ are unorthodox and do not follow the normal standards. I suggest that the notations are changed to " $\delta^{30}\text{Si}$ of ASi" and " $\delta^{30}\text{Si}$ of DSi", respectively.

101 and 104: Insert number of samples in the brackets so that readers do not have to search the entire text for this important information.

259: Delete the second "record".

275-276: This statement must be supported by a reference.

Reviewer #3 (Remarks to the Author):

I have reviewed the revised version of this manuscript. The authors have addressed all of my concerns, but I remain skeptical of the model. I recognize the improvements to the modeling regarding including a third box to capture the influence of high latitude dynamics. I remain concerned regarding its simplicity. Qualitatively the trends are likely correct, but the quantitative details of timing and magnitude have high uncertainty. This is inherent to the model's simplicity that no level of bootstrapping can overcome. The manuscript does present the details of the modeling clearly and in sufficient detail that readers can judge the output for themselves, provided they have the expertise for critical evaluation. As a first-order pass the new model is a fair effort.

Reviewer #2 (Remarks to the Author):

Major comments

I was reviewer #2 in the last round of reviewing. In my opinion, the authors have done a good job in addressing my comments and improving the manuscript. In particular, I prefer the new modeling exercise. It is still my opinion that the riverine $\delta^{30}\text{Si}$ data is interesting and worth publishing, but I have to admit that I am still skeptical to the study design.

- 1. I was critical to the premise of the study that “until recently, it was assumed that riverine $\delta^{30}\text{Si}$ input was uniform over glacial-interglacial timescales” (38-39). The authors removed this postulate from the Abstract but chose to keep it in the main text (38-39). In their response, the authors highlight studies that examine $\delta^{30}\text{Si}$ input, but they still fail to show who “assumed that $\delta^{30}\text{Si}$ input was uniform” and the statement remains unsupported by literature references. Just because trends in a flux or a composition have not been investigated, it cannot be concluded that the understanding in the research community is or was that fluxes and compositions have not varied over time. In fact, I will claim that the general understanding is that riverine ion fluxes and isotope compositions (e.g., $\delta^{87}\text{Li}$, $\delta^{26}\text{Mg}$, $\delta^{44}\text{Ca}$, $\delta^{56}\text{Fe}$) varied from LGM to today because observations from natural low-temperature environments today show that these fluxes and compositions vary (of course I cannot support my claim with literature references). I hope that the authors now better understand my argumentation and my recommendation to rephrase this sentence.*

We have rephrased this sentence to reflect the reviewer’s concern. However, we maintain that the general assumption among the isotope community (especially regarding Si isotopes) until present was that inputs did not change substantially. For example, some of the first interpretations of $\delta^{30}\text{Si}$ in sediment cores didn’t consider any change in input composition (Beucher et al., 2007; De La Rocha et al., 1998; Horn et al., 2011). We have edited the text to further highlight the potential of glacial contributions to the marine silica cycle.

- 2. I still find the study design strange, and the authors have not revised the manuscript to clarify their thoughts behind selecting a field site in west Greenland and a marine core off southern Iceland to the readers. For a study published in a Nature journal, I would expect researchers to optimize the study design to provide the best results possible, although I of course respect the logistical and economical costs that can be financed by a research proposal. In its current form, the study seems opportunistic.*

As per our response in the first response to reviewers, we agree that, ideally, the glacial meltwater and core samples would be located closer together. However, to emphasise, collection of core data is a **major** undertaking, and we are instead making use of previously collected materials that are very well characterised. First, there are numerous logistical and economical challenges that would be involved with collecting a core in this region of the Davis Strait (where no well-characterised, high-quality, high-resolution cores currently exist that do not have radiocarbon age reversals (Andrews et al., 2014)). Such an undertaking is extremely expensive and beyond the scope of current funding constraints. High-resolution contourite deposit cores have been recovered from sites on Erik Drift, but these sites will also be influenced by the Irminger Current, rather than waters originating from the West Greenland Ice Sheet. Second, the Iceland core provided an opportunity to retrieve a high-resolution archive of spicules from thick sediment deposits over the appropriate time period, in order to test the hypothesis we present in the manuscript. It's essential when working with novel isotope systems to work on good quality cores that have already been worked on (and characterised) extensively using more traditional isotope proxies, which is also why this core was selected for the study. We have given a full justification in terms of knowing that the core site - proximal to the extent of the ice sheet - represents overflow water influence, in the manuscript. We detail below why using a catchment on basaltic rock would not be ideal for reconstructing palaeo-flux estimates.

- 3. The authors have chosen to keep the statement that "Leverett Glacier ... is an order of magnitude larger than other glaciers studied for Si to data", and in the rebuttal letter the authors restate that "Leverett Glacier is at least an order of magnitude larger in size than other glaciers studied in this context to date". First, the sentence refers to "glaciers studied for Si", not glaciers studied for both "Si concentrations and $\delta^{30}\text{Si}$ composition", as the authors seem to indicate in their response to my comment.*

We have changed the wording of this sentence to address the reviewers concern.

4. *Secondly, I was actually thinking of Skeidararjökull, which would be the obvious choice for a field site, as this outlet glacier is located close to the offshore core site and drains towards the south. The catchment area of Skeidararjökull is generally estimated to approximately 1380 km², or more than twice the area of the glacier used in this study. In addition to published spot sampling, Gíslason et al. (Chem. Geol. 190, 2002) presented a time-series of Si concentrations and Opfelgelt et al. (2013) presented a $\delta^{30}\text{Si}$ value.*

We maintain that this is the first study of an ablation season at a glacier to monitor in detail the dissolved, amorphous and corresponding $\delta^{30}\text{Si}$ of meltwaters from a large glacial catchment. The Gíslason et al (2002) manuscript mentioned by the reviewer concerned 20 samples taken from outburst flood waters following a subglacial volcanic eruption in 1996. Although this is a very interesting paper, it details meltwater chemistry during a single volcanic event, that is not during the normal state of the river. We do not believe these samples and a single spot measurement constitute an ablation season time series. Furthermore, Gíslason et al did not measure and/or document amorphous silica concentrations.

Leverett Glacier arguably provides a much better analogue to palaeo ice sheet meltwaters at large, as we document and explain in the manuscript. The large palaeo ice sheets that covered much of North America and northern Europe were underlain by Precambrian shield bedrock, not basalts. Furthermore, Leverett Glacier has been well studied, and is therefore well characterised. Hydrologically active catchment area, discharge data, and associated electrochemical variables are also well monitored. It therefore provides an ideal setting to study large glacial catchment systems. We maintain this is the case here.

Finally, and importantly, the core record is designed to demonstrate the potential impact of incoming meltwaters on the water column Si cycle. It does not require the use of Icelandic meltwater values to do this, as the timings of meltwater pulse events are concurrent across the northern hemisphere. The interpretation of this high resolution sediment core would not change even if Icelandic values were used.

5. *Thirdly, it is not obvious for readers to know when the authors think that a study qualifies as a study “studied for Si” if a presentation of Si concentrations is not enough. How should readers know that a study that “has a combined discharge of three glacial systems” (rebuttal letter) or studies from the Himalayas (e.g., Batura or Gangotri glaciers) or studies from Antarctica are disqualified? The sentence does not mention anything about that the studies must be comparable to Leverett Glacier in all aspects.*

As Skeidararjökull is significantly larger than the Greenlandic field site, I am confused by some of the authors’ responses such as: “We would also like to reiterate a point made previously that the size of glacial catchment sampled also appears important in dictating geochemical composition of waters (Wadham et al., 2010), therefore sampling smaller glaciers in Iceland may also not be representative of the larger glacial catchments that would have existed with the Icelandic Ice Sheet”. I would have liked to see Si and $\delta^{30}\text{Si}$ time-series from Skeidararjökull included in the study in combination with time-series from a granitic site such as the one from Greenland. After all, this is submission to a Nature journal.

The reviewer has misunderstood our point here. There are clearly other studies that include data on dissolved silica. We do not dispute this - indeed many of the authors involved in this study have been involved in other such studies. Our main point is that these do either do not cover similar periods of time or only include information on dissolved silica concentrations. This study is novel because it incorporates a comprehensive dataset (both hydrological and geochemical) from a large, well characterised ice sheet catchment.

We do not know how the catchment area of Skeidararjökull was calculated, and are not disputing the reviewer’s quoted catchment area, but there is no referenced paper which details this figure, how it was calculated, and whether it is the hydrologically active catchment or the whole glacial catchment. Leverett Glacier is well studied, and the hydrologically active part of the catchment is comparatively well constrained. For example, if we calculated Leverett Glacier catchment area by topography, it has been estimated as $>1,200 \text{ km}^2$, yet the hydrologically active drainage area is $\sim 600 \text{ km}^2$. Similarly, Skeidararjökull may very well have a topographical drainage area of $\sim 1,300 \text{ km}^2$, but the hydrologically active drainage area might be substantially less. We can find no reference to these details in the literature. Furthermore, there is no seasonal data (except that of a volcanic outburst event) and no amorphous silica data from Skeidararjökull. We repeat our assertions above that the

large palaeo ice sheets that covered much of North America and northern Europe were underlain by Precambrian shield bedrock, not basalts, therefore using Skeidararjökull would not make sense for studying wholesale deglaciation.

We do not fully understand what the reviewer refers to as a “Si and $\delta^{30}\text{Si}$ time-series from Skeidararjökull”, but assume they mean that we collect a time series of data from this glacier as well. This is unfeasible and would involve a huge amount of logistical effort, and funding to carry out properly, and is far outside the scope of this study. The reason these data are rare is because long time series data in these environments is challenging to collect. Not only are high quality discharge and hydrochemical measurements difficult to obtain and maintain, but the person power needed to run a remote field camp for 3 months cannot be overstated. We hope this study will provide further impetus to collect such data.

We have now included specific reference to the silicon isotope value attained from Skeidararjökull in the manuscript, along with approximate dissolved silica concentrations. We cannot use the silica data from Gíslason et al (2002), as suggested, because this was such an extreme event – the dissolved silica concentrations are an order of magnitude higher than those recorded in Opfergelt et al. (2013), which we assume to be normal flow. We would like to highlight that even with these data more explicitly referenced in the manuscript, our message and interpretation of the marine sediment core $\delta^{30}\text{Si}$ record does not change. See lines 155-156 and 277-280 for changes.

Other comments:

42, 46: Palaeo- is a prefix. Insert hyphens.

Done

91: Thanks for pointing my attention to Hawkings et al. (2017). However, the authors oversell the representativeness of Si fluxes from Leverett Glacier in 2015 here. As I understand Hawkings et al. (2017), “previously reported for the GrIS” actually refers to “previously reported for Leverett Glacier in 2012”, as the comparison is only between fluxes in runoff.

This information is clearly stated in the text – see lines 96-97 “...within the same range as those reported at Leverett Glacier for the 2009-2012 period...”. ASi has not been measured

anywhere else, and our values lie within those reported by Meire et al. (2016). We have now added this reference to the first sentence of this section.

91-92: The readers should know more about the interannual variations at Leverett Glacier. Annual differences in DSi concentrations of 9.6 and 20.8 μM seem large and need to be presented for the reader and discussed in more detail in relation to the model assumptions. As above, our values reported here are within previously reported ranges for the GrIS. The differences we observe at Leverett Glacier are incorporated into the sensitivity testing of the model (given that we use the full range of values reported in this study). Moreover, any difference in the DSi concentration is far outweighed by the easily dissolvable ASi phase (again a minimum and maximum estimate is used here). This difference in DSi concentration therefore has relatively little impact on the outputs of the modelling component of this study.

100, 104, elsewhere: The notations $\delta^{30}\text{ASi}$ and $\delta^{30}\text{DSi}$ are unorthodox and do not follow the normal standards. I suggest that the notations are changed to " $\delta^{30}\text{Si}$ of ASi" and " $\delta^{30}\text{Si}$ of DSi", respectively.

We believe the way that we presented this data makes it clear immediately to the reader which phase we are referring to in a manner that cannot be misread or misinterpreted, even if they have not previously been standard notations. We also acknowledge the reviewer's concern and have provided an explicit definition to these terms in the manuscript. We have made what these notations refer to clearer on lines 103 and 107.

101 and 104: Insert number of samples in the brackets so that readers do not have to search the entire text for this important information.

Done.

259: Delete the second "record".

Done.

275-276: This statement must be supported by a reference.

Reference added.

Reviewer #3 (Remarks to the Author):

Major comment

I have reviewed the revised version of this manuscript. The authors have addressed all of my concerns, but I remain skeptical of the model. I recognize the improvements to the modeling regarding including a third box to capture the influence of high latitude dynamics. I remain concerned regarding its simplicity. Qualitatively the trends are likely correct, but the quantitative details of timing and magnitude have high uncertainty. This is inherent to the model's simplicity that no level of bootstrapping can overcome. The manuscript does present the details of the modeling clearly and in sufficient detail that readers can judge the output for themselves, provided they have the expertise for critical evaluation. As a first-order pass the new model is a fair effort.

We would like to emphasise that this is not foremost a modelling paper. The model exercise in this manuscript is intended to demonstrate that the magnitude of changes in Si flux and $\delta^{30}\text{Si}$ implied by our dataset are significant relative to the ocean's Si inventory and $\delta^{30}\text{Si}$. For this primary purpose, estimating the magnitudes and timescales of ocean Si changes, the use of a box model is clearly adequate and the results are robust. With regard to the parameterisation of biological Si cycling, the referee is absolutely right to state that the model is too simplistic to infer/predict the biogeochemical and ecosystem response to variable Si input. We have rewritten parts of the main text to make very clear that our model thought experiment targets the marine Si inventory changes by isolating Si input as the only model forcing (e.g., see lines 113-117, and 213-216). When addressing the biological, ecological and carbon cycle implications of the simulated Si inventory changes we are careful to reference relevant literature rather than rely on dynamics that arise in our model.

With regard to the referee's comment on uncertainties, the uncertainty in timing and magnitude in our model ensemble comes from both the (partially model-based) reconstruction of input fluxes and from the extrapolation from Leverett Glacier data, and this is acknowledged in the manuscript. The individual ensemble member simulations encapsulate these sources of uncertainty. The ocean Si inventory and $\delta^{30}\text{Si}$ respond to all ensemble members because the timescale of deglaciation is similar to the ocean's Si residence time and the implied fractional changes in Si input are significant. That is, as long as we're approximately right with the oceanic residence time of Si over the deglaciation (and the model runs were constructed so that this should be the case), the response of the ocean (in

terms of timescale and magnitude) is governed by mass balance. Because deglacial changes in Si input to the ocean operate on two timescales (i.e. LGM to present versus centennial melt-water pulses) the explicit model calculations add value to the manuscript because they illustrate the relative impacts of long-term and short term changes and because they contextualise our sponge spicule $\delta^{30}\text{Si}$ record.

References cited

Andrews, J.T., Gibb, O.T., Jennings, A.E., Simon, Q. (2014) Variations in the provenance of sediment from ice sheets surrounding Baffin Bay during MIS 2 and 3 and export to the Labrador Shelf Sea: site HU2008029-0008 Davis Strait. *Journal of Quaternary Science* 29, 3-13.

Beucher, C.P., Brzezinski, M.A., Crosta, X. (2007) Silicic acid dynamics in the glacial sub-Antarctic: Implications for the silicic acid leakage hypothesis. *Global Biogeochemical Cycles* 21.

De La Rocha, C.L., Brzezinski, M.A., DeNiro, M.J., Shemesh, A. (1998) Silicon-isotope composition of diatoms as an indicator of past oceanic change. *Nature* 395, 680-683.

Gíslason, S.R., Snorrason, Á., Kristmannsdóttir, H.K., Sveinbjörnsdóttir, Á.E., Torsander, P., Ólafsson, J., Castet, S., Dupré, B. (2002) Effects of volcanic eruptions on the CO₂ content of the atmosphere and the oceans: the 1996 eruption and flood within the Vatnajökull Glacier, Iceland. *Chemical Geology* 190, 181-205.

Horn, M.G., Beucher, C.P., Robinson, R.S., Brzezinski, M.A. (2011) Southern ocean nitrogen and silicon dynamics during the last deglaciation. *Earth and Planetary Science Letters* 310, 334-339.

Meire, L., Meire, P., Struyf, E., Krawczyk, D.W., Arendt, K.E., Yde, J.C., Juul Pedersen, T., Hopwood, M.J., Rysgaard, S., Meysman, F.J.R. (2016) High export of dissolved silica from the Greenland Ice Sheet. *Geophysical Research Letters* 43, 9173-9182.

REVIEWERS' COMMENTS:

Reviewer #2 (Remarks to the Author):

During the last round of reviewing, the authors have made minor adjustments to the text to address most of my previous concerns, although the authors maintain most of their viewpoints. As I also maintain most of my critic, we simply disagree on a few issues. However, these issues are minor and they should not prevent the publication of this manuscript. It is still my opinion that the riverine $\delta^{30}\text{Si}$ time-series is very interesting and worthy of publication, the model is simple but useful, but I am not a huge fan of the comparison between the model results based on the Greenlandic data and the results from the Icelandic marine core. Overall, the manuscript is in a publishable state but I would prefer if the authors considered my comments below before a potential publication.

35-42: The authors "maintain that the general assumption among the isotope community (especially regarding Si isotopes) until present was that inputs did not change substantially" (quote from their rebuttal letter), while I maintain my impression that for more than a decade it has been the general hypothesis that riverine $\delta^{30}\text{Si}$ inputs to oceans have varied substantially on glacial-interglacial time-scales. To my knowledge, Georg et al. (2006) was the key paper advocating for variable riverine $\delta^{30}\text{Si}$ to oceans on glacial-interglacial time-scales. Here is a quote from the Abstract: "The presence of seasonal variations in Si isotope composition in mountainous rivers provides evidence that extreme changes in climate affect the overall composition of dissolved Si delivered to the oceans. The oceanic Si isotope composition is very sensitive to even small changes in the riverine Si isotope composition and this parameter appears to be more critical than plausible changes in the Si flux. Therefore, concurrent changes in weathering style may need to be considered when using the Si isotopic compositions of diatoms, sponges and radiolarian as paleoproductivity proxies", and another quote from section 4.5 Glacial-interglacial variations in seawater $\delta^{30}\text{Si}$: "our results indicate that the $\delta^{30}\text{Si}$ composition of rivers can undergo large variations that relate to weathering style. Small offsets in average riverine $\delta^{30}\text{Si}$ from +1.0‰ to +1.2‰ and +0.8‰, result in the average continental input changing from +0.8‰ to +0.98‰ and +0.64‰, respectively (Table 3), and thus significantly change the Si isotopic input into the oceans". It is my strong belief that since the publication of Georg et al. in 2006 (more than a decade ago), it has been the generally accepted hypothesis that riverine $\delta^{30}\text{Si}$ inputs to oceans have varied substantially on glacial-interglacial time-scales. I cannot recall any papers or conference presentations that have argued against Georg et al. (2006). It is correct that researchers have assumed constant $\delta^{30}\text{Si}$ values as river inputs to oceans in models, but model assumptions do not necessarily reflect the general understanding of spatial and temporal variations in variables. It will almost be similar to writing that "Until 2018, studies have assumed an absence of any changes in marine Si cycling and constant aeolian dust fluxes from LGM to present day" and then cite this manuscript, knowing that this is a model simplification of the actual understanding. This is a manuscript submitted to a Nature journal, and as such I, and hopefully many other readers, have high expectations of being presented the full story. I do not believe that the full story of the state-of-the-art is presented in the Introduction. Therefore, I strongly encourage the authors to reconsider their viewpoint and rephrase lines 35-42.

42 and throughout the manuscript: Please delete the space after palaeo-. A hyphen is missing in line 47.

256-258 and 280-283: It is relevant to add one or two sentences about the origin of meltwater during MWP1a (and probably also MWP2). MWP1a happened most likely during the stadial Oldest Dryas and was triggered by ice sheet collapse and a positive feedback between eustatic sea level rise and extreme iceberg calving (e.g. Norðdahl and Ingólfsson 2015). Therefore, the enhanced meltwater and Si fluxes during MWP1a derived very likely from melting of icebergs rather than from riverine meltwater runoff. It would be interesting for readers to know, what effects enhanced Si from icebergs may have on the model input data (e.g. the use of riverine Si data in the model)

and on the model results. Also, it has been debated whether the main source of MWP1a was the Laurentide ice sheet or the Antarctic ice sheet (e.g., Peltier 2005).

277: Please use Icelandic letters in the spelling of Skeiðarárjökull.

277-280: I am glad to see that the authors have included a sentence about Si data from Iceland. In their rebuttal letter, the authors note that they could not find any references for the catchment area of Skeiðarárjökull and in the text they write ">1000 km²". In the literature, the catchment area of Skeiðarárjökull has been estimated between ca. 1300 km² (Björnsson 2017, page 406) and ca. 1500 km² (Tweed et al. 2005). As far as I remember, Randolph Glacier Inventory 5.0 uses a catchment area of 1428 km², which is similar to the catchment area delineation by Aðalsgeirsdóttir (2003, Table 7.1, page 81). Matthew Roberts uses a catchment area of 1370-1380 km² in his work. It is very inaccurate to write ">1000 km²", so I recommend that the authors change this value to a more precise estimate of the catchment area. Skeiðarárjökull is a temperate glacier (e.g., Tweed et al. 2005), so it is by definition hydrologically active beneath its entire catchment area.

298: Are the references (1,13,59) correct? This should be references to papers, which not fully appreciate the part played by ice sheets and glaciers in marine nutrient and nutrient isotope cycling. In my opinion, references #1 and #13 appreciate the contribution by ice sheets and glaciers. It is also not clear to me, why reference #59 is cited in this context.

309-311: Delete the last part of this sentence: "..., making fundamental revisions to our interpretation of palaeoceanographic proxies and our understanding of past and present carbon cycling". This is a huge exaggeration of the scientific impact of the findings in the study. Instead, I will suggest something like "..., supporting the hypothesis by Georg et al. (2006) that ..." or simply delete the sentence.

320: Six references are not needed to guide interested readers to details regarding the gauging of Leverett Glacier. A single reference to the study with most details on the gauging is enough. The rest of the references are superfluous.

Figure 4 caption, 686: Insert references to the data sources of the glacial and non-glacial runoff. Also, explain the shaded regions in the caption.

Figures 4 and 5: Use uppercase letters in the figures.

Figures 4, 5 and 6: The paper mainly focuses on MWP1a and MWP2, but these three figures show model input and output data and Si isotope data back to 26,000 year BP and 23,000 year BP. However, the consequences of deposition of enhanced ice-rafted detritus (IRD) during Heinrich event 1 (H1; 15,000 to 18,000 year BP) have not be account for. Although H1 had limited influence on sea level rise as it is mainly related to the break-up of ice shelves, it may have had a significant impact on DSi and ASi fluxes to the North Atlantic and to the Si isotope compositions. If the authors agree with me on this, then it should to be mentioned in the text that the model does not account for the enhanced IRD flux during H1, so that readers do not misinterpret model results for H1 in Figures 4, 5 and 6.

References:

Aðalsgeirsdóttir, G. Flow dynamics of Vatnajökull ice cap, Iceland. PhD thesis, ETH Zürich (2003).

Björnsson, H. The Glaciers of Iceland. Atlantis Press, Paris. ISBN 978-94-6239-206-9 (2017).

Georg, R.B., Reynolds, B.C., Frank, M., Halliday, A.N. Mechanisms controlling the silicon isotopic compositions in river waters. *Earth and Planetary Science Letters*, 249(3-4), 290-306 (2006).

Norðdahl, H., Ingólfsson, Ó. Collapse of the Icelandic ice sheet controlled by sea level rise? *arktos* 1, 1-18 (2015).

Peltier, W.R. On the hemispheric origins of meltwater pulse 1a. *Quaternary Science Reviews*, 24, 1655-1671 (2005).

Tweed, F.S., Roberts, M.J., Russell, A.J. Hydrological monitoring of supercooled meltwater from Icelandic glaciers. *Quaternary Science Reviews*, 24, 2308-2318 (2005).

Response to Reviewer

We once again thank the reviewer for their constructive comments on the manuscript. In our revised manuscript and reviewer responses below we hope to have addressed the remaining concerns.

Reviewer 2

During the last round of reviewing, the authors have made minor adjustments to the text to address most of my previous concerns, although the authors maintain most of their viewpoints. As I also maintain most of my critic, we simply disagree on a few issues. However, these issues are minor and they should not prevent the publication of this manuscript. It is still my opinion that the riverine $\delta^{30}\text{Si}$ time-series is very interesting and worthy of publication, the model is simple but useful, but I am not a huge fan of the comparison between the model results based on the Greenlandic data and the results from the Icelandic marine core. Overall, the manuscript is in a publishable state but I would prefer if the authors considered my comments below before a potential publication.

We thank the reviewer for reviewing the manuscript again and are delighted that they feel the manuscript is appropriate for publication. We agree that there are some minor points we simply disagree on, one being the location of the marine sediment core and our field site, and we do not think this should prevent publication of the manuscript. We hope to have addressed the remaining concerns below.

35-42: The authors “maintain that the general assumption among the isotope community (especially regarding Si isotopes) until present was that inputs did not change substantially” (quote from their rebuttal letter), while I maintain my impression that for more than a decade it has been the general hypothesis that riverine $\delta^{30}\text{Si}$ inputs to oceans have varied substantially on glacial-interglacial time-scales. To my knowledge, Georg et al. (2006) was the key paper advocating for variable riverine $\delta^{30}\text{Si}$ to oceans on glacial-interglacial time-scales. Here is a quote from the Abstract: “The presence of seasonal variations in Si isotope composition in mountainous rivers provides evidence that extreme changes in climate affect the overall composition of dissolved Si delivered to the oceans. The oceanic Si isotope composition is very sensitive to even small changes in the riverine Si isotope composition and this parameter appears to be more critical than plausible changes in the Si flux.

Therefore, concurrent changes in weathering style may need to be considered when using the Si isotopic compositions of diatoms, sponges and radiolarian as paleoproductivity proxies”, and another quote from section 4.5 Glacial-interglacial variations in seawater $\delta^{30}\text{Si}$: “our results indicate that the $\delta^{30}\text{Si}$ composition of rivers can undergo large variations that relate to weathering style. Small offsets in average riverine $\delta^{30}\text{Si}$ from +1.0‰ to +1.2‰ and +0.8‰, result in the average continental input changing from +0.8‰ to +0.98‰ and +0.64‰, respectively (Table 3), and thus significantly change the Si isotopic input into the oceans”. It is my strong belief that since the publication of Georg et al. in 2006 (more than a decade ago), it has been the generally accepted hypothesis that riverine $\delta^{30}\text{Si}$ inputs to oceans have varied substantially on glacial-interglacial time-scales. I cannot recall any papers or conference presentations that have argued against Georg et al. (2006). It is correct that researchers have assumed constant $\delta^{30}\text{Si}$ values as river inputs to oceans in models, but model assumptions do not necessarily reflect the general understanding of spatial and temporal variations in variables. It will almost be similar to writing that “Until 2018, studies have assumed an absence of any changes in marine Si cycling and constant aeolian dust fluxes from LGM to present day” and then cite this manuscript, knowing that this is a model simplification of the actual understanding. This is a manuscript submitted to a Nature journal, and as such I, and hopefully many other readers, have high expectations of being presented the full story. I do not believe that the full story of the state-of-the-art is presented in the Introduction. Therefore, I strongly encourage the authors to reconsider their viewpoint and rephrase lines 35-42.

We thank the reviewer for pointing out this study and agree that these findings should be highlighted in our study. We have made changes to the Introduction and added this reference in response to these comments. Please see lines 38-43.

42 and throughout the manuscript: Please delete the space after palaeo-. A hyphen is missing in line 47.

Done.

256-258 and 280-283: It is relevant to add one or two sentences about the origin of meltwater during MWP1a (and probably also MWP2). MWP1a happened most likely during the stadial Oldest Dryas and was triggered by ice sheet collapse and a positive feedback between eustatic sea level rise and extreme iceberg calving (e.g. Norðdahl and Ingólfsson 2015).

Therefore, the enhanced meltwater and Si fluxes during MWP1a derived very likely from melting of icebergs rather than from riverine meltwater runoff. It would be interesting for readers to know, what effects enhanced Si from icebergs may have on the model input data (e.g. the use of riverine Si data in the model) and on the model results. Also, it has been debated whether the main source of MWP1a was the Laurentide ice sheet or the Antarctic ice sheet (e.g., Peltier 2005).

The timing of MWP1a is uncertain and by MWP2 we assume the reviewer is referring to MWP1b. The latest sea level constraints place it around the time of the Bolling Warming ~14.5 ka (Deschamps et al., 2012), but these data do not preclude the possibility of it happening several hundred years either side of the abrupt Northern Hemisphere warming event (e.g. ~14 ka, during the Older Dryas; see discussion in Ivanovic et al., 2017). The study referred to by the reviewer concerns the Icelandic ice sheet, which is relatively very small ($3.29 \times 10^5 \text{ km}^3$; Norðdahl and Ingólfsson 2015) compared to other major ice caps (North America/Laurentide, Antarctica, Eurasia and Greenland ice sheets), containing only ~0.7% of the global volume of terrestrial ice at the Last Glacial Maximum (Velichko et al., 1997; Peltier et al., 2015; Ivanovic et al., 2016). Whilst iceberg calving from the Icelandic ice cap may have contributed towards MWP1a, it certainly cannot explain the recorded 12-22 m global mean sea level rise.

Although an Antarctic versus Laurentide source of MWP1a has historically been debated, there is now a growing consensus in the literature that points to a mixed-origin of MWP1a (e.g. Gomez et al., 2015; Liu et al., 2016), likely with around half coming from the interior of the Laurentide ice sheet (~5-8 m; Gregoire et al., 2012, 2016) and smaller contributions from Antarctica (likely < 2m; Golledge et al., 2014), Eurasia (~2.5 m; Patton et al. (2017)) and Greenland (~0.5 m). Thus, runoff from melting terrestrial ice makes up the dominant proportion of MWP1a, explaining at least 10 m sea level rise in <350 years, and there is no evidence that iceberg calving was sufficient to contribute anywhere near as much meltwater/sea level rise as this. Our extensive discussion of Si borne by terrestrial-runoff analogous to Laurentide-runoff (which explains around half of MWP1a) is therefore much more appropriate to this manuscript than iceberg-derived Si contributions.

We have added some text to lines 297-308 to make this clearer.

277: Please use Icelandic letters in the spelling of Skeiðarárjökull.

Done

277-280: I am glad to see that the authors have included a sentence about Si data from Iceland. In their rebuttal letter, the authors note that they could not find any references for the catchment area of Skeiðarárjökull and in the text they write “>1000 km²”. In the literature, the catchment area of Skeiðarárjökull has been estimated between ca. 1300 km² (Björnsson 2017, page 406) and ca. 1500 km² (Tweed et al. 2005). As far as I remember, Randolph Glacier Inventory 5.0 uses a catchment area of 1428 km², which is similar to the catchment area delineation by Aðalsgeirsdóttir (2003, Table 7.1, page 81). Matthew Roberts uses a catchment area of 1370-1380 km² in his work. It is very inaccurate to write “>1000 km²”, so I recommend that the authors change this value to a more precise estimate of the catchment area. Skeiðarárjökull is a temperate glacier (e.g., Tweed et al. 2005), so it is by definition hydrologically active beneath its entire catchment area.

We have made changes to the text here to address the reviewer’s concern. The term “hydrologically active” in this context does not refer to the region of the bed which is wet (and thus the debate between warm and cold bedded ice). The reference cited refers instead to the portion of the glacier that contributes surface melt and therefore feeds subglacial hydrological evolution over a melt season, as calculated by a degree day model (Cowton et al., 2012 – cited in the text). The topographically defined catchment area of Leverett Glacier is ~1,200 km² (Palmer et al., 2011) similar to the size of Skeiðarárjökull), which includes the accumulation zone of the glacier.

298: Are the references (1,13,59) correct? This should be references to papers, which not fully appreciate the part played by ice sheets and glaciers in marine nutrient and nutrient isotope cycling. In my opinion, references #1 and #13 appreciate the contribution by ice sheets and glaciers. It is also not clear to me, why reference #59 is cited in this context. *Reference 1 does not include any mention of glaciers or ice sheets in the evaluation of the marine Si cycling, even when discussion potential silica inputs. We have changed the wording of this sentence to reflect the reviewers concern.*

309-311: Delete the last part of this sentence: “..., making fundamental revisions to our interpretation of palaeoceanographic proxies and our understanding of past and present carbon cycling”. This is a huge exaggeration of the scientific impact of the findings in the study. Instead, I will suggest something like “..., supporting the hypothesis by Georg et al.

(2006) that ...” or simply delete the sentence.

We have toned down the wording of this sentence.

320: Six references are not needed to guide interested readers to details regarding the gauging of Leverett Glacier. A single reference to the study with most details on the gauging is enough. The rest of the references are superfluous.

These were added in response to a comment from a previous reviewer comments. None of these references are additional to the text and point the reader toward studies where this technique to used.

Figure 4 caption, 686: Insert references to the data sources of the glacial and non-glacial runoff. Also, explain the shaded regions in the caption.

These sources are described in detail in the supplementary information. We have added some explanation of the shaded regions in the revised manuscript.

Figures 4 and 5: Use uppercase letters in the figures.

Done.

Figures 4, 5 and 6: The paper mainly focuses on MWP1a and MWP2, but these three figures show model input and output data and Si isotope data back to 26,000 year BP and 23,000 year BP. However, the consequences of deposition of enhanced ice-rafted detritus (IRD) during Heinrich event 1 (H1; 15,000 to 18,000 year BP) have not be account for. Although H1 had limited influence on sea level rise as it is mainly related to the break-up of ice shelves, it may have had a significant impact on DSi and ASi fluxes to the North Atlantic and to the Si isotope compositions. If the authors agree with me on this, then it should to be mentioned in the text that the model does not account for the enhanced IRD flux during H1, so that readers do not misinterpret model results for H1 in Figures 4, 5 and 6.

The reviewer is correct in that we do not include the fluxes of IRD during H1, even though they may be substantial sources of DSi and ASi to the open ocean. There are two reasons for not including these in our current estimates. First, the freshwater fluxes from Heinrich events are extremely poorly constrained. Second, at present no $\delta^{30}\text{Si}$ for iceberg ASi and DSi exist. The potential impact of Heinrich events would therefore be mostly guesswork. We hope this manuscript will lead to further interest in this topic. We have added some text to the Methodology (lines 415-416) and Discussion (lines 297-307) to address this comment.

References cited

- Deschamps, P., Durand, N., Bard, E., Hamelin, B., Camoin, G., Thomas, A.L., Henderson, G.M., Okuno, J., Yokoyama, Y. (2012) Ice-sheet collapse and sea-level rise at the Bolling warming 14,600 years ago. *Nature* 483, 559-564.
- Golledge, N.R., Menviel, L., Carter, L., Fogwill, C.J., England, M.H., Cortese, G., Levy, R.H. (2014) Antarctic contribution to meltwater pulse 1A from reduced Southern Ocean overturning. *Nature Communications* 5, 5107.
- Gomez, N., Gregoire, L.J., Mitrovica, J.X., Payne, A.J. (2015) Laurentide-Cordilleran Ice Sheet saddle collapse as a contribution to meltwater pulse 1A. *Geophysical Research Letters* 42, 3954-3962.
- Gregoire, L.J., Otto-Bliesner, B., Valdes, P.J., Ivanovic, R. (2016) Abrupt Bølling warming and ice saddle collapse contributions to the Meltwater Pulse 1a rapid sea level rise. *Geophysical Research Letters* 43, 9130-9137.
- Gregoire, L.J., Payne, A.J., Valdes, P.J. (2012) Deglacial rapid sea level rises caused by ice-sheet saddle collapses. *Nature* 487, 219.
- Ivanovic, R.F., Gregoire, L.J., Kageyama, M., Roche, D.M., Valdes, P.J., Burke, A., Drummond, R., Peltier, W.R., Tarasov, L. (2016) Transient climate simulations of the deglaciation 21–9 thousand years before present (version 1) – PMIP4 Core experiment design and boundary conditions. *Geosci. Model Dev.* 9, 2563-2587.
- Ivanovic, R.F., Gregoire, L.J., Wickert, A.D., Valdes, P.J., Burke, A. (2017) Collapse of the North American ice saddle 14,500 years ago caused widespread cooling and reduced ocean overturning circulation. *Geophysical Research Letters* 44, 383-392.
- Liu, J., Milne, G.A., Kopp, R.E., Clark, P.U., Shennan, I. (2015) Sea-level constraints on the amplitude and source distribution of Meltwater Pulse 1A. *Nature Geoscience* 9, 130.
- Norðdahl, H., Ingólfsson, Ó. (2015) Collapse of the Icelandic ice sheet controlled by sea-level rise? *arktos* 1, 1-18.

Palmer, S., Shepherd, A., Nienow, P., Joughin, I. (2011) Seasonal speedup of the Greenland Ice Sheet linked to routing of surface water. *Earth and Planetary Science Letters* 302, 423-428.

Patton, H., Hubbard, A., Andreassen, K., Auriac, A., Whitehouse, P.L., Stroeven, A.P., Shackleton, C., Winsborrow, M., Heyman, J., Hall, A.M. (2017) Deglaciation of the Eurasian ice sheet complex. *Quaternary Science Reviews* 169, 148-172.

Peltier, W.R., Argus, D.F., Drummond, R. (2015) Space geodesy constrains ice age terminal deglaciation: The global ICE-6G_C (VM5a) model. *Journal of Geophysical Research-Solid Earth* 120, 450-487.

Velichko, A.A., Kononov, Y.M., Faustova, M.A. (1997) The last glaciation of earth: Size and volume of ice-sheets. *Quaternary International* 41-42, 43-51.